# Label-free biosensor assay decodes the dynamics of Toll-like receptor signaling

Janine Holze[1], Felicitas Lauber[1], Sofía Soler [2], Evi Kostenis [3] &
Günther Weindl [1]✉

The discovery of Toll-like receptors (TLRs) represented a significant break-through that paved the way for the study of host-pathogen interactions in innate immunity. However, there are still major gaps in understanding TLR function, especially regarding the early dynamics of downstream TLR pathways. Here, we present a label-free optical biosensor-based assay as a method for detecting TLR activation in a native and label-free environment and defining the dynamics of TLR pathway activation. This technology is sufficiently sensitive to detect TLR signaling and readily discriminates between different TLR signaling pathways. We define pharmacological modulators of cell surface and endosomal TLRs and downstream signaling molecules and uncover TLR signaling signatures, including potential biased receptor signaling. These findings highlight that optical biosensor assays complement traditional assays that use a single endpoint and have the potential to facilitate the future design of selective drugs targeting TLRs and their downstream effector cascades.

Toll-like receptors (TLRs) form the first barrier in the innate immune response and therefore represent a highly interesting and therapeutically promising target to modulate immune response and inflammation[1,2]. The TLR family consists of 10 members (TLR1-TLR10) in humans and 12 members (TLR1-TLR9, TLR11-TLR13) in mice and is known to detect a variety of pathogen-associated molecular patterns (PAMPs) of invading microorganisms (e.g., lipids, lipoproteins, proteins, and nucleic acids) that enable the host to distinguish between different infections[3]. In addition to PAMPs, also stress- or injury-released molecules, termed damage- or danger-associated molecular patterns, can be sensed by TLRs. They can be broadly classified into two subfamilies based on their localization: cell surface TLRs (TLR1, 2, 4-6, 10) and intracellular TLRs (TLR3, 7-9, 11-13). Mechanistically, TLRs dimerize as homo- or heterodimers for activation, and trigger intracellular pathways that promote diverse cellular responses, including inflammatory processes[4]. Dysregulated TLR signaling has been implicated in various inflammatory and autoimmune diseases, including sepsis, rheumatoid arthritis, cancer, metabolic disorders, and

neurodegenerative diseases. Understanding the roles of TLRs in these conditions is essential for developing targeted therapies that can modulate immune responses, leading to improved treatment outcomes. Most of our understanding on TLR signaling comes from loss-of-function genetic analyses and modulation of TLRs by small-molecule ligands. These studies typically focus on a limited range of downstream signaling pathways and effector proteins. However, commonly used measurements, such as cytokine release and protein phosphorylation, often provide delayed insights. Their regulation by multiple transcription factors and other proteins complicates real-time evaluation of activation. Interpreting these downstream responses is difficult because they occur long after the initial TLR activation events and are susceptible to signal amplification. Similarly, investigations frequently use fluorescent tags or reporter systems, which, while well-characterized and invaluable for experimentation, require careful and time-consuming validation. This is essential to prevent potential interference with the natural protein structure, function, or the proper activation of signaling pathways. Given these limitations, innovative

[1]Pharmaceutical Institute, Section Pharmacology and Toxicology, University of Bonn, Bonn, Germany. [2]Institute of Experimental Haematology and Transfusion Medicine, University Hospital Bonn, Bonn, Germany. [3]Institute for Pharmaceutical Biology, Molecular, Cellular and Pharmacobiology Section, University of Bonn, Bonn, Germany. ✉e-mail: guenther.weindl@uni-bonn.de

approaches that allow a holistic view on TLR signaling and its modulation are highly desired[5].

In G protein coupled receptor (GPCR) research, the concept of analyzing whole cell responses—rather than individual components of signaling pathways—has been well established[6]. The approach using label-free whole-cell biosensing, for example based on dynamic mass redistribution (DMR), not only offers access to rich cellular information but also provides exciting mechanistic insights into GPCR activation and ligand-dependent modulation[7–9]. This technology uses a polarized broadband light that passes through the bottom of a biosensor microtiter plate containing the cells. Upon receptor activation, a shift in the wavelength of the reflected light reveals signal-transduction events linked to changes in cell morphology[10]. Depending on the signaling pathways activated and the cell-shaped change-related mass redistribution, the direction (positive or negative) and magnitude of the detected shift in wavelength may vary (Fig. 1a).

Optical biosensor assays have the main advantage in analyzing living cells in real time and identifying signal events by detecting whole-cell responses related to changes in cellular shape[11]. These morphological changes can be observed as long as the signaling event results in a rearrangement of the cytoskeleton[6,12]. Unlike traditional assays that use a single endpoint, DMR offers the possibility of detecting the cumulative response of living cells by high-throughput screening. This opens up opportunities for exploring and defining pathway-selective effector activation or biased signaling of TLRs. The latter represents the ability of a ligand to selectively activate some but not all signaling pathways and is a well-established paradigm for GPCRs[13,14]. The TLR family is also known to activate multiple signaling pathways; however, their bias signaling capabilities are not yet well-defined, and our understanding of TLR ligand bias is less advanced compared to our knowledge of GPCR ligand bias[15,16].

The chemical toolbox for studying the dynamics of TLR signaling has steadily increased[17], however, label-free approaches are still very limited. Here, we demonstrate that label-free detection captures receptor activation and signaling of membrane (TLR1, 2, 4, 6) and endosomal (TLR3, 8) TLRs in real time, which has not, to the best of our knowledge, been reported by other technology platforms. Our data provide evidence for distinct mechanisms of TLR pathway activation that occur immediately after receptor engagement, along with potential ligand bias, offering valuable insights into TLR signal transduction.

## Results

### Label-free optical biosensor assay discriminates TLR4 signaling of LPS chemotypes

TLR4 is activated by lipopolysaccharide (LPS), a key component of the outer membrane of Gram-negative bacteria[18]. LPS consists of lipid A, a core oligosaccharide, and an O side chain[19] (Supplementary Fig. 1). In particular, the hydrophobic component, lipid A, is responsible for its immune stimulating activity. The structures of lipid A in various bacteria display variability in the number or length of attached fatty acids, a critical factor for TLR4 activation. This activation of the TLR4 subtype by LPS is known to initiate the MyD88 signaling pathway at the plasma membrane, as well as CD14-dependent endocytosis, which initiates the Toll/interleukin 1 receptor domain-containing adapter inducing interferon β (TRIF, also known as TICAM-1)-dependent cascade[20–22] (Fig. 1b). To assess whether LPS-induced signals can be detected by the optical biosensor, we used HEK293 cells, which endogenously express TRIF-related adaptor molecule (TRAM, also known as TICAM-2) and TRIF (Supplementary Fig. 2), and were stably transfected to express TLR4, MD-2 and CD14. The cells were stimulated with increasing concentrations of the TLR4 ligand LPS Escherichia coli (LPS E. coli) and DMR signals were recorded (Fig. 1c). No signals were detected in the presence of the TLR4 antagonist TAK-242 (Fig. 1e) or control HEK293 cells lacking TLR4 (Fig. 1f), demonstrating that the LPS-induced DMR signal originates specifically from TLR4 activation. The TLR4 inhibitor TAK-242 alone induced no detectable signal (Fig. 1g). HEK293 TLR4/MD-2/CD14 and control reporter cells lacking TLR4 (HEK293 Null2) showed characteristic DMR signals of G protein signaling[6,23] in the presence of the GPCR ligands acetylcholine, epinephrine, iperoxo, or atropine, respectively. This illustrates that cells overexpressing TLR4 show consistent GPCR signaling patterns compared to control cells lacking TLR4 (Supplementary Fig. 3a, b). LPS from E. coli concentration-dependently induced an early negative peak at 12 min followed by a large continuing positive change in DMR signals for concentrations above 10 ng/ml (Fig. 1c). The DMR response distinctly varied for LPS from Salmonella minnesota (LPS S. minnesota), displaying an early and robust positive signal starting at 25 min. For both LPS chemotypes, positive signals did not return to baseline during the recording period. In contrast, LPS from Rhodobacter sphaeroides (LPS R. sphaeroides), a TLR4 antagonist[24], induced no detectable signal (Fig. 1d).

To gain a deeper understanding of the pharmacology related to DMR signals induced by the different LPS and to study the kinetics of receptor potency ($EC_{50}$) and efficacy ($E_{max}$) at early signaling events, we established full concentration-effect curves generated by using six different time points (Fig. 1h). We found a time-dependent increase in potency and efficacy for LPS from E. coli that remained constant after 117 min (Table 1). The efficacy increased after 50 min and remained unchanged throughout the recording time. For LPS from S. minnesota we observed a similar time-dependent change in potency, whereas the efficacy decreased after 50 min. Collectively, these data indicate that DMR readily discriminates between LPS of different bacteria and uncovers a differential concentration- and chemotype-dependent signaling signature.

To confirm that the changes induced by LPS affect cytoskeletal function and lead to morphological alterations, we preincubated cells with inhibitors of actin or tubulin polymerization. Since inhibiting actin- or tubulin-dependent cytoskeletal restructuring could cause cell detachment, appearing as a loss of mass, the experiment was conducted in suspension mode rather than adherent mode. Sufficient time was allowed for baseline equilibration before adding LPS E. coli (Supplementary Fig. 4a–c). To verify that the LPS E. coli signal detected in suspension HEK293 TLR4/MD-2/CD14 reporter cells is mediated by TLR4, the inhibitor TAK-242 was used (Fig. 2a). Both actin inhibitors, cytochalasin B[25] and latrunculin A[26], as well as the microtubule inhibitor nocodazole[27], reduced the LPS E. coli-mediated response in a concentration-dependent manner (Fig. 2b–d). Throughout the detection period, cells remained viable in the presence of the inhibitors (Supplementary Fig. 4d).

### Optical biosensor assay uncovers differential signaling of TLR2 heterodimers

TLR2 operates as a heterodimer with TLR1 or TLR6[28]. TLR2/1 and TLR2/6 heterodimers are known to initiate similar signaling cascades, resulting in comparable gene activation profiles[29]. However, we and others have reported variations in signaling and sensitivity to ligands for different TLR2 heterodimers[30–35]. Furthermore, agonists for TLR2/1 and TLR2/6 induce NF-κB, MAP kinase activation, and cytokine transcription with different kinetics[36]. To assess whether TLR2 activation and signal variances are also detectable with whole-cell biosensing, we focused on TLR2 (Fig. 3a). To ensure specific activation of the receptor, we used the agonists $Pam_3CSK_4$ (preferentially TLR2/1) and $Pam_2CSK_4$ (preferentially TLR2/6), and HEK293 reporter cells that are stably transfected with only one of the heterodimers after knock-out of endogenous TLR1 or TLR6, respectively[35]. Consistent with our previous findings, $Pam_3CSK_4$ predominantly activated TLR2/1, while $Pam_2CSK_4$ mainly activated TLR2/6 in a concentration-dependent manner (Fig. 3b). However, both $Pam_2CSK_4$ and $Pam_3CSK_4$ were also able to activate the other dimer, although to a lesser extent (Supplementary

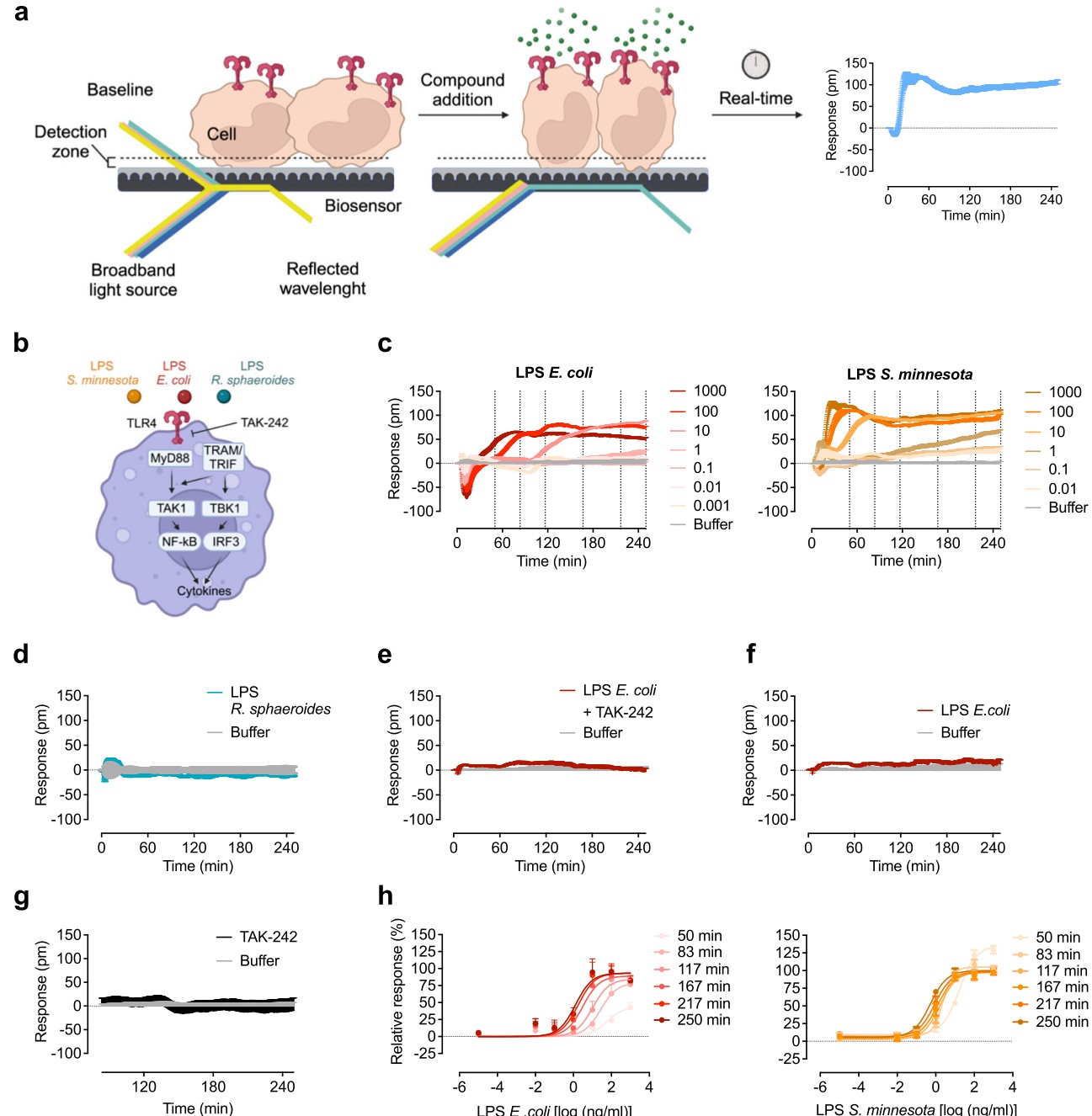

**Fig. 1 | Optical biosensor discriminates different LPS chemotypes in HEK293 TLR4/MD-2/CD14 reporter cells. a** DMR principle: cultured cells are grown in 384 well microplates equipped with a resonant waveguide grating biosensor within the bottom of each well. A specific broadband light source illuminates the lower surface of the biosensor. The illumination generates an energy field within the DMR detection zone, and its strength diminishes exponentially with the distance from the sensor. Under baseline conditions (left) cells are in close proximity to the biosensor, and the refractive index of these cells determines the reflected wavelength, which is subsequently measured. If cells undergo morphological changes (middle), the refractive indices above the sensor can either increase or decrease. Consequently, the reflected wavelength becomes shorter or longer, respectively. The shift in the measured wavelength is plotted against the recording time (right). Schematic illustration adapted from ref. 42) **b** Schematic representation of TLR4 signaling and contact point of TAK-242. **c, d** HEK293 TLR4/MD-2/CD14 reporter cells were stimulated with the indicated concentrations (ng/ml) of LPS

from *E. coli, S. minnesota* and *R. sphaeroides* (1000 ng/ml). Dashed lines represent the six time points that were used to generate concentration-effect curves (**h**). **e** HEK293 TLR4/MD-2/CD14 reporter cells were preincubated with 50 μM of the TLR4 antagonist TAK-242 or (**f**) HEK293 control reporter cells lacking TLR4 (Null2) were stimulated with LPS *E. coli*. (1000 ng/ml). **g** HEK293 TLR4/MD-2/CD14 reporter cells were incubated with TAK-242 (50 μM). Baseline-corrected DMR recordings are mean + SEM and representative of three biologically independent experiments. **h** Sigmoidal concentration effect curves resulting from DMR traces (**c**) of *n* biologically independent experiments (**h**: *n* = 4 biological replicates except LPS *E. coli* 50 min *n* = 3 (Mean ± SEM)). Concentration-effect curves of DMR data were generated by the response at six different time points. Calculated pharmacological parameters of the concentration-effect curves are depicted in Table 1. Source data are provided as a Source Data file. (**a**) and (**b**) was created in BioRender. Weindl, G. (2024) **a**: BioRender.com/k68r667 **b**: BioRender.com/j94y620.

**Table 1 | Pharmacological parameter of LPS induced DMR at six selected time points in HEK293 TLR4/MD-2/CD14 reporter cells**

| | Time (min) | Bottom (%) | Top (%) | $n_H$ | logEC$_{50}$ | $n$ |
|---|---|---|---|---|---|---|
| LPS *E. coli* | 50 | =0 | 44 ± 7[#(0.0144)] | =1.00 | 1.77 ± 0.29 | 3 |
| | 83 | =0 | 80 ± 7 | =1.00 | 1.43 ± 0.18 | 4 |
| | 117 | =0 | 85 ± 7 | =1.00 | 1.01 ± 0.18[§(0.0498)] | 4 |
| | 167 | =0 | 89 ± 7 | =1.00 | 0.50 ± 0.21[*(0.0093)] | 4 |
| | 217 | =0 | 93 ± 7 | =1.00 | 0.26 ± 0.21[*(0.0019)§(0.0095)] | 4 |
| | 250 | =0 | 93 ± 7 | =1.00 | 0.11 ± 0.21[*(0.0007)§(0.0033)] | 4 |
| LPS *S. minnesota* | 50 | 10 ± 3[$(<0.0001)] | 135 ± 5[#(0.006)] | =1.00 | 1.20 ± 0.10 | 4 |
| | 83 | =0 | 105 ± 4 | 1.66 | 0.38 ± 0.11[*(<0.0001)] | 4 |
| | 117 | =0 | 99 ± 4 | =1.00 | 0.17 ± 0.10[*(<0.0001)] | 4 |
| | 167 | =0 | 97 ± 3 | =1.00 | 0.08 ± 0.08[*(<0.0001)] | 4 |
| | 217 | =0 | 98 ± 3 | =1.00 | −0.29 ± 0.07[*(<0.0001)§(0.0007)&(0.0226)] | 4 |
| | 250 | =0 | 100 ± 2 | =1.00 | −0.25 ± 0.08[*(0.0001)§(0.0014)&(0.0425)] | 4 |

Curve fitting of single experiments was obtained by nonlinear regression analysis applying a three ($n_H$, Hill coefficient =1.00) or four-parameter logistic equation. Values are means ± SEM.
[#]significantly different from 100%, [$]significantly different from 0% (one-sample *t*-test). [*]significantly different from values at 50 min. [§]significantly different from values at 83 min. [&]significantly different from values at 117 min, one-way analysis of variance (ANOVA, Tukey's post-test).

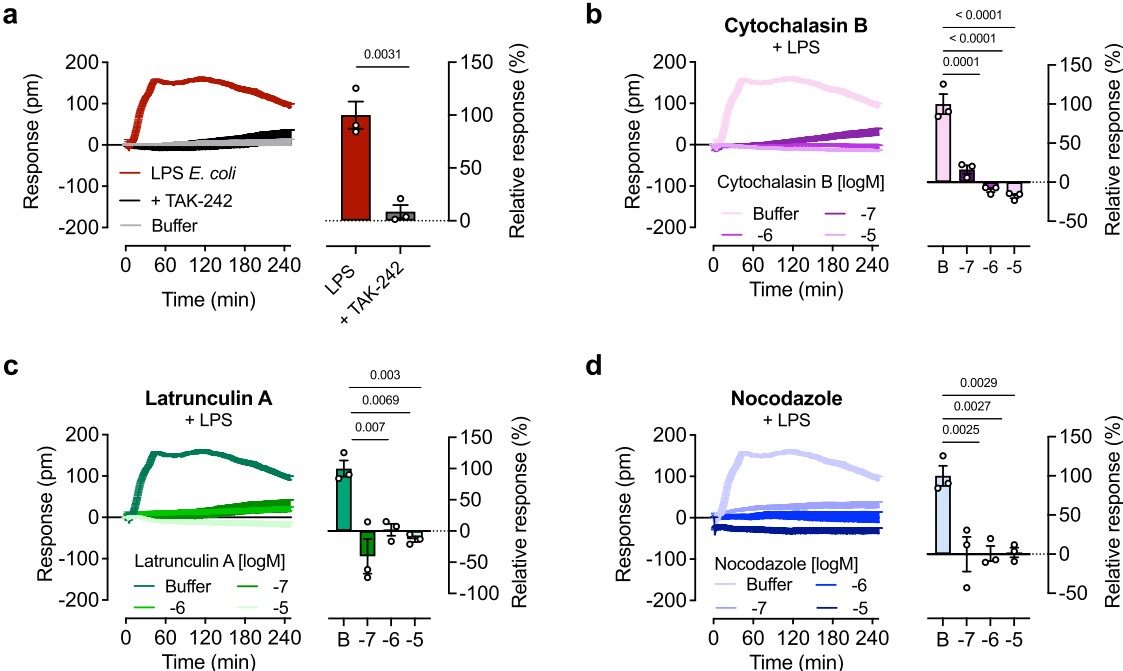

**Fig. 2 | Inhibition of actin and tubulin network abolishes LPS induced DMR changes. a** HEK293 TLR4/MD-2/CD14 reporter cells in suspension were stimulated with the indicated concentrations (ng/ml) of LPS from *E. coli* with or without 50 µM of the TLR4-antagonist TAK-242. HEK293 TLR4/MD-2/CD14 reporter cells were preincubated with the indicated concentrations of actin and microtubule inhibitors (**b**) cytochalasin B, (**c**) latrunculin A or (**d**) nocodazole stimulated with LPS *E. coli* (1000 ng/ml). Baseline-corrected DMR recordings are mean + SEM and representative of three biologically independent experiments. (**a**–**d**, right panels) Values at 250 min are presented as mean + SEM and are normalized to LPS *E. coli* (1000 ng/ml) (*n* = 3 biologically independent experiments). Two-tailed Student's *t* test (**a**) and one-way analysis of variance (ANOVA, Tukey's post-test) (**b**–**d**). Source data are provided as a Source Data file.

Fig. 5a). No signals were detected in HEK293 TLR4/MD-2/CD14 reporter cells (Supplementary Fig. 5b) or control HEK293 cells lacking TLR1 and 6 (Supplementary Fig. 5c). The latter cells also showed characteristic DMR signals of G protein signaling in the presence of acetylcholine or epinephrine, respectively (Supplementary Fig. 5d). Similar to LPS, stimulation with Pam$_3$CSK$_4$ and Pam$_2$CSK$_4$ resulted in a positive DMR signal that persisted throughout the recording period. Both TLR2 agonists showed distinct DMR profiles, with Pam$_3$CSK$_4$ inducing a faster early decline followed by an increase compared to Pam$_2$CSK$_4$. The concentration-effect curves (Fig. 3c) revealed a different kinetic dependence compared to LPS. The efficacy of Pam$_3$CSK$_4$ decreased more pronounced and gradually over time (Table 2), while its potency

remained constant. In comparison, the TLR2/6 agonist Pam$_2$CSK$_4$ demonstrated constant efficacy, but the potency decreased significantly after 217 min. These findings offer evidence to suggest that both agonists trigger differences in the recruitment of adaptor proteins, including MyD88 (Myeloid differentiation primary response 88) and Mal (Myd88 adaptor-like protein)[36] as well as signal cascade kinetics. To gain further insights into the signaling traces induced by the TLR2 heterodimers, we used HEK293 TLR2 reporter cells, which endogenously express both dimer partners TLR1 and TLR6[37,38]. Both agonists induce similar traces as shown in cells expressing only one heterodimer (Fig. 3d). Next, we tested two TLR2 antagonists which differ in inhibitory potency and heterodimer predominance[35]

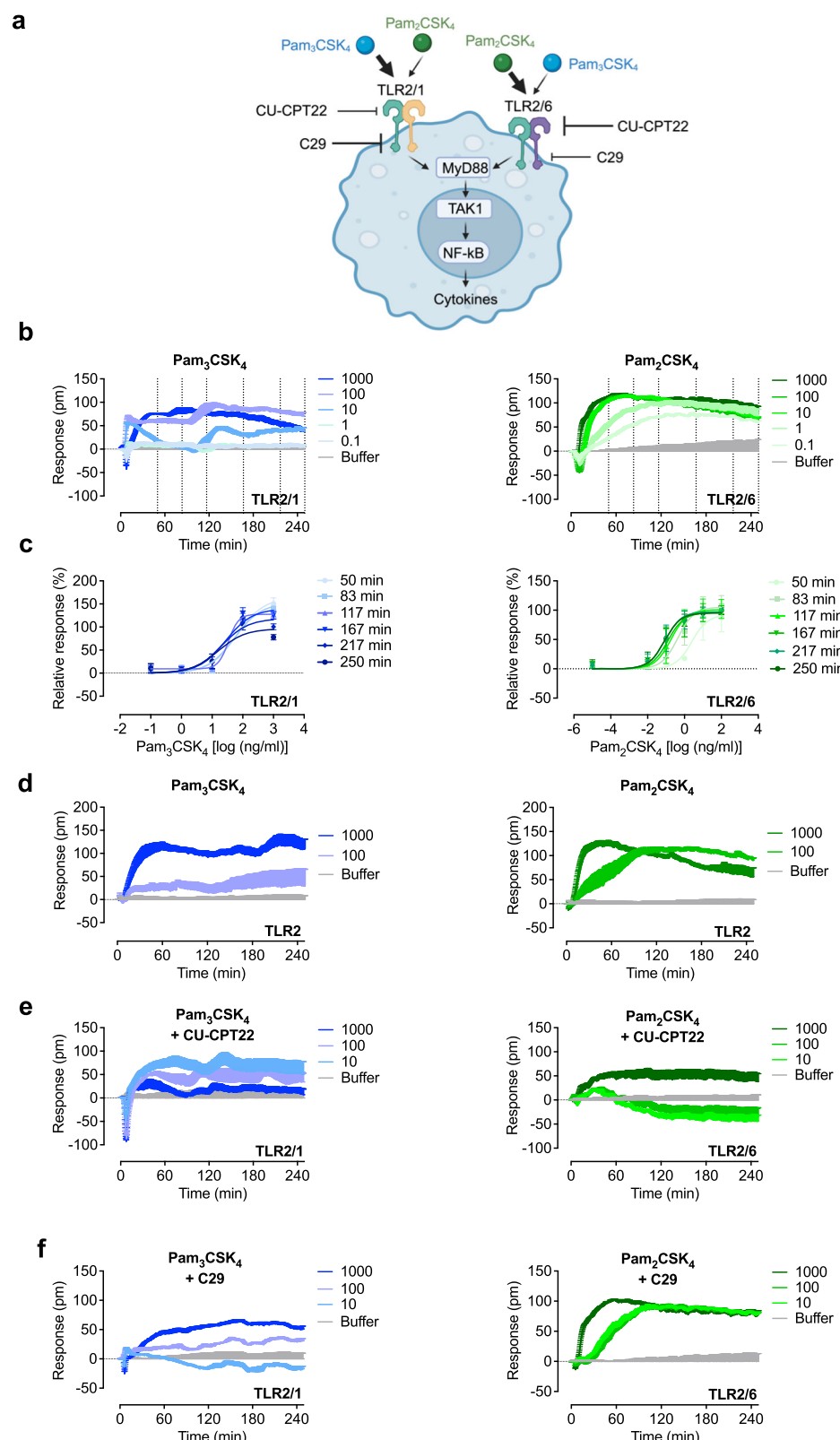

(Fig. 3a). We confirmed that CU-CPT22 preferentially inhibits Pam₂CSK₄ signaling, whereas C29 shows only weak effects for both heterodimers (Fig. 3e, f). Together, our data demonstrate the detectability of TLR2 and subsequent pathway activation. Furthermore, we have shown that heterodimerization of TLR2 with TLR1 or TLR6 leads to clearly distinct optical signals, indicating differential signaling behavior.

**Optical biosensor assay detects TLR subtypes in native cell lines**
Label-free biosensor assays offer the possibility of investigating receptor activation in a native cell setting[39]. We tested whether the technology can be applied to cell lines of different origin that express endogenous TLRs. We used the well-characterized epidermal HaCaT cell line that responds to TLR2 ligands, but shows no functional TLR4 signaling[40]. As expected, stimulation with LPS from *E. coli* generated a

**Fig. 3 | Optical biosensor assays detect differential signaling of TLR2 heterodimers. a** Schematic representation of TLR2 signaling, ligand priority and contact point of the two antagonists CU-CPT22 and C29. **b** HEK293 TLR2/1 or 2/6 reporter cells were stimulated with the indicated concentrations (ng/ml) of $Pam_3CSK_4$ or $Pam_2CSK_4$. Dashed lines represent the six time points that were used to generate concentration-effect curves (**c**). Baseline-corrected DMR recordings are mean + SEM and representative of three biologically independent experiments. **c** Sigmoidal concentration-effect curves resulting from DMR traces of *n* biologically independent experiments (**c**: *n* = 3 biological replicates except $Pam_3CSK_4$ log 3, log 2, log 1 *n* = 5, $Pam_2CSK_4$ log 0, 50 min *n* = 2) (Mean ± SEM). Concentration-effect curves of

DMR data were generated by the response at six different time points (**b**). **d** HEK293 TLR2 reporter cells were stimulated with the indicated concentrations (ng/ml) of $Pam_3CSK_4$ or $Pam_2CSK_4$. HEK293 TLR2/1 or 2/6 reporter cells were preincubated with 50 μM of the TLR2 antagonist (**e**) CU-CPT22 or (**f**) C29 stimulated with the indicated concentrations (ng/ml) of $Pam_3CSK_4$ or $Pam_2CSK_4$. Calculated pharmacological parameters of the concentration-effect curves (**c**) are depicted in Table 2. Baseline-corrected DMR recordings are mean + SEM and representative of three biologically independent experiments. Source data are provided as a Source Data file. (**a**) was created in BioRender. Weindl, G. (2024) BioRender.com/q83w265.

**Table 2 | Pharmacological parameters of $Pam_3CSK_4$- and $Pam_2CSK_4$-induced DMR at six selected time points in HEK293 TLR2/1 and TLR2/6 reporter cells**

| | Time (min) | Bottom (%) | Top (%) | $n_H$ | $logEC_{50}$ | *n* |
|---|---|---|---|---|---|---|
| **TLR2/1 $Pam_3CSK_4$** | 50 | =0 | 163 ± 11[#(0.0046)] | =1.00 | 1.80 ± 0.12 | 5 |
| | 83 | =0 | 152 ± 9[#(0.0045)] | =1.00 | 1.72 ± 0.11 | 5 |
| | 117 | =0 | 128 ± 8[#(0.0249)] | 2.17 ± 0.76 | 1.42 ± 0.16 | 5 |
| | 167 | =0 | 140 ± 12[#(0.029)] | =1.00 | 1.44 ± 0.18 | 5 |
| | 217 | =0 | 118 ± 12 | =1.00 | 1.30 ± 0.22 | 5 |
| | 250 | =0 | 96 ± 11[*(0.002)§(0.0117)] | =1.00 | 1.20 ± 0.26 | 5 |
| **TLR2/6 $Pam_2CSK_4$** | 50 | =0 | 88 ± 14 | =1.00 | 0.39 ± 0.34 | 3 |
| | 83 | =0 | 105 ± 12 | =1.00 | −0.36 ± 0.30 | 3 |
| | 117 | =0 | 101 ± 11 | =1.00 | −0.73 ± 0.29 | 3 |
| | 167 | =0 | 96 ± 10 | =1.00 | −0.90 ± 0.27 | 3 |
| | 217 | =0 | 97 ± 7 | =1.00 | −1,05 ± 0.21[*(0.0259)] | 3 |
| | 250 | =0 | 95 ± 6 | =1.00 | −1.08 ± 0.19[*(0.0227)] | 3 |

Curve fitting of single experiments was obtained by nonlinear regression analysis applying a three ($n_H$, Hill coefficient =1.00) parameter logistic equation. Values are means ± SEM. [#]significantly different from 100% (one-sample *t*-test). [*]significantly different from values at 50 min. [§]significantly different from values at 83 min, one-way analysis of variance (ANOVA, Tukey's post-test).

signal identical to control, while $Pam_3CSK_4$ and $Pam_2CSK_4$ concentration-dependently induced DMR responses (Fig. 4a and Supplementary Fig. 6a). The signal generated by $Pam_3CSK_4$ was nearly indistinguishable from that seen in HEK293 cells expressing TLR2/1 (Figs. 3b and 4b). However, the inherent cellular phenotype should be considered when interpreting the data obtained in both cell lines. The DMR signal for $Pam_2CSK_4$ was different in HaCaT cells and TLR2/6 reporter cells. HaCaT cells lack the expression of TLR6[40] and $Pam_2CSK_4$ can activate TLR2/1 heterodimers[35,41], therefore we reasoned that the TLR2/6 ligand triggers TLR2/1 in HaCaT cells. Indeed, the signal for $Pam_2CSK_4$ obtained in TLR2/1 reporter cells was close to that detected in HaCaT cells and clearly differed from $Pam_3CSK_4$ (Fig. 3b and Supplementary Fig. 6a). In the presence of the TLR2 antagonists CU-CPT22 and C-29, the $Pam_3CSK_4$-induced responses were inhibited (Fig. 4c). Both antagonists induced initial transient signals that returned to baseline before stimulation with $Pam_3CSK_4$ (Supplementary Fig. 6b). Overlapping signals can be excluded for C29. However, since CU-CPT22 induced negative signals at later time points, interference with $Pam_3CSK_4$-induced signals may have occurred in HaCaT cells, independent of TLR2 inhibition. The optical traces triggered by the lipopeptides were largely independent of TLR4 (Supplementary Fig. 6c). The $Pam_2CSK_4$-induced signals shifted to more negative values in the presence of TAK-242, without altering the overall course of the signals. For control purposes, we confirmed that acetylcholine and epinephrine, which act via endogenous GPCRs, and the adenylyl cyclase-activating agent forskolin showed the signal trace fingerprints typical for HaCaT cells[42] (Supplementary Fig. 6d).

### Label-free optical biosensor assay reveals TLR ligand bias

LPS from different sources triggers distinct inflammatory responses and shows a form of biased signaling[16]. To test whether label-free, cell-based assays using optical biosensor technology can reliably study TLR signaling in cells expressing endogenous levels of TLRs and detect

potential biased signaling, we used THP-1 cells, which can activate both TLR4 signaling pathways. The monocytic cell line grows in suspension but can be induced to differentiate into adherent macrophage-like cells. In line with the poor response of THP-1 monocytes to LPS from *E. coli*[43], only a weak negative signal was observed in undifferentiated THP-1 cells (Fig. 5a, left) while PMA-differentiated cells responded to *E. coli* LPS with robust signals (Fig. 5b, left). LPS from *S. minnesota*, on the other hand, could elicit a detectable signal in both THP-1 monocytes and macrophages (Fig. 5a, b, right). The different ability to induce strong signals in THP-1 monocytes can be attributed to differences in the characteristics of LPS. LPS from *E. coli* is classified as smooth (s)-form LPS, while LPS from *S. minnesota* is categorized as rough (r)-form LPS[44]. Although both share the same receptor complex, their mechanisms of action differ. CD14 is critical for NF-κB und IRF signaling triggered by sLPS, but it is not required for the rLPS-induced pathway[45]. Given that THP-1 monocytes express little[46] to no CD14 (Supplementary Fig. 7) and were stimulated under serum-free conditions, our findings are in line with these observations. To confirm our results, we generated THP-1 TLR4-KO cells, which were unable to induce cytokine expression in response to LPS *E. coli* but retained the ability to respond to the TLR2/1 ligand $Pam_3CSK_4$ (Supplementary Fig. 8). No DMR signals were detected after incubation with LPS *E. coli* and LPS *S. minnesota*, confirming that the signals detected in THP-1 cells are mediated by TLR4 (Fig. 5c, d). To further investigate the signals in THP-1 cells and the role of CD14, we used THP1-Dual TLR4/MD-2/CD14 cells (hereafter referred to as THP-1 Dual cells). These cells stably express CD14 and we expect them to produce signals similar to those in THP-1 macrophages. LPS *E. coli* and LPS *S. minnesota* induced signals in THP-1 Dual cells nearly identical to those observed in THP-1 macrophages (Fig. 5e), which also express CD14 (Supplementary Fig. 7). No traces could be recorded in THP1-Dual KO-TLR4/MD-2/CD14 cells (hereafter referred to as THP-1 Dual TLR4-KO cells) (Fig. 5f) and with LPS *R. sphaeroides* (Supplementary Fig. 9a, b), whereas TLR2-

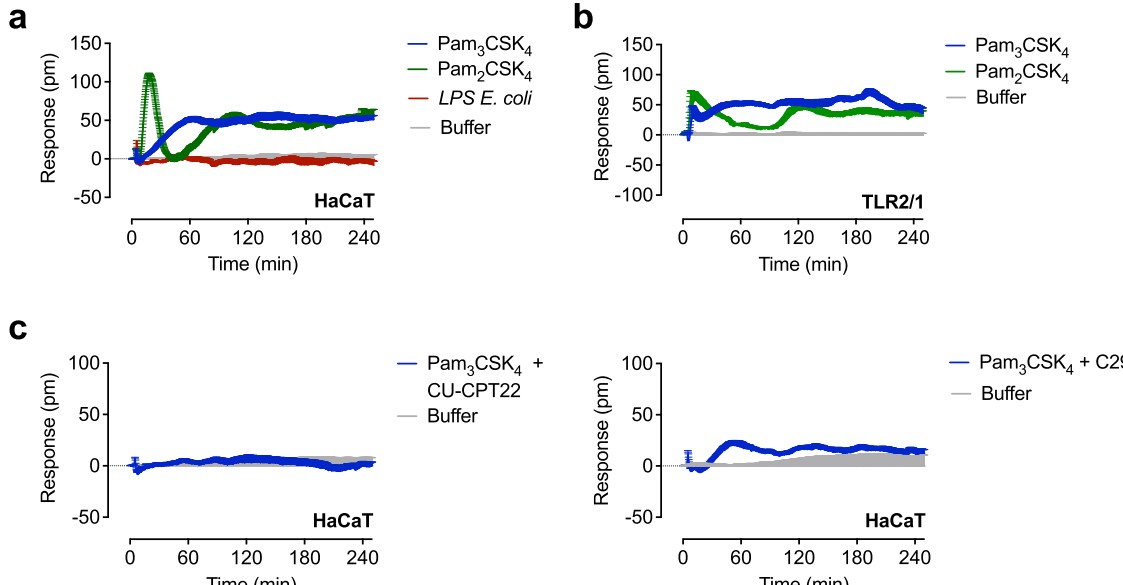

**Fig. 4 | Optical biosensor uncovers TLR2 and TLR4 signaling in epidermal HaCaT cells. a** HaCaT cells were stimulated with LPS *E. coli* (1000 ng/ml), Pam$_3$CSK$_4$ (1000 ng/ml) or Pam$_2$CSK$_4$ (1000 ng/ml). **b** HEK293 TLR2/1 reporter cells were stimulated with Pam$_3$CSK$_4$ (100 ng/ml) or Pam$_2$CSK$_4$ (100 ng/ml). **c** HaCaT cells were preincubated with 50 μM of the TLR2 antagonists CU-CPT22 or C29 stimulated with Pam$_3$CSK$_4$ (1000 ng/ml). Baseline-corrected DMR recordings are mean + SEM and representative of three biologically independent experiments. Source data are provided as a Source Data file.

induced responses remained intact (Supplementary Fig. 9c, d). The obtained DMR signals corresponded with NF-κB (Supplementary Fig. 9e, g) and IRF activity (Supplementary Fig. 9 f, h).

Both in THP-1 macrophages as well as in THP-1 Dual cells, LPS chemotypes induce unique signal fingerprints at early time points. Thus, we analyzed whether the different signals lead to differential transcriptional responses. We performed RNA sequencing of THP-1 macrophages after 3 h stimulation with LPS from *E. coli* and *S. minnesota*, respectively. For control purposes, we used HEK293 TLR4/MD-2/CD14 cells. In both cell lines, the global transcriptional response induced by the LPS chemotypes was similar (Fig. 5g, h). As expected, in HEK293 TLR4/MD-2/CD14 cells, the LPS chemotypes induced a weak transcriptional response with 52 genes differentially expressed compared to control (Fig. 5i), whereas a strong response with 2559 differentially expressed genes was observed in THP-1 macrophages (FDR adjusted *p*-value < 0.05 and | logFC|> 1) (Fig. 5j). *E. coli* LPS specifically modulated 16 and 605 genes in TLR4-HEK293 cells and THP-1 macrophages, respectively. In contrast, around twofold less genes were specifically regulated by LPS from *S. minnesota*. However, the volcano plots showed no significant differences between both LPS chemotypes, except for one gene in HEK293 TLR4/MD-2/CD14 cells (Supplementary Fig. 10a, b).

We next stimulated THP-1 monocytes with the TLR2 ligands Pam$_3$CSK$_4$ and Pam$_2$CSK$_4$ and found early positive signals that increased over time reaching a plateau at 45 min and 30 min, respectively (Fig. 5k). We observed a marked difference in the optical trace signatures in THP-1 macrophages (Fig. 5l). The signals for Pam$_3$CSK$_4$ and Pam$_2$CSK$_4$ clearly resembled oscillatory dynamics with a fast initial peak at 12 and 10 min, a negative peak at 35 and 30 min followed by a second positive peak at 85 and 80 min before finally reaching a second negative peak at 180 and 175 min, respectively.

Using HEK293 cells expressing specific TLRs, as well as native cell lines such as HaCaT cells and THP-1 monocytes and macrophages, we have demonstrated that agonist-induced signals can be differentiated by label-free technology. Next, we sought to study TLR dynamics in primary monocytes isolated from peripheral blood mononuclear cells (PBMCs). As expected, nontoxic concentrations of the tested TLR4 and TLR2 agonists induced cytokine secretion (Supplementary Fig. 11a, b). LPS from *E. coli* and *S. minnesota* displayed concentration-dependent

positive signals (Fig. 6a) which were inhibited in the presence of TAK-242 (Fig. 6b). In line with the results in cell lines, *S. minnesota* LPS showed a faster signal increase compared to *E. coli* LPS. The TLR2/6 agonist Pam$_2$CSK$_4$ induced concentration-dependent signals, whereas the TLR2/1 agonist Pam$_3$CSK$_4$ showed signals only at the highest concentration (Fig. 6c). Our findings demonstrate that TLR2 and TLR4 signaling differs between cell types and shows unique optical signatures.

## LPS chemotypes differentially induce MyD88-dependent signaling

TLR4 induces both MyD88-dependent and -independent pathways[47,48]. MyD88 triggers rapid NF-κB activation and the release of proinflammatory cytokines such as tumor necrosis factor-alpha (TNF), interleukin (IL-)1β, IL-6 and IL-8. The MyD88-independent pathway leads to rapid activation of interferon regulatory factor (IRF)3, resulting in the release of interferon (IFN)-β and delayed NF-κB activation[49]. Based on the differences identified by the optical biosensor in the early response to LPS from *E. coli* and *S. minnesota*, we hypothesized that the LPS chemotypes differentially activate MyD88-dependent and -independent pathways. We used the MyD88 inhibitor ST2825, a synthetic compound that mimics the structure of the heptapeptide in the BB-loop of the MyD88-TIR domain, thereby interfering with the homodimerization of MyD88, a crucial step for pathway activation[50] (Fig. 7a). No signals were detected in the presence of ST2825 (Supplementary Fig. 12). ST2825 completely abolished the signal induced by *E. coli* LPS in HEK293 TLR4/MD-2/CD14 cells during the recording period (Fig. 7b). After stimulation with LPS from *S. minnesota*, ST2825 substantially reduced the signal until 40 min, however, the signal gradually increased and remained at around 60% compared to cells stimulated with LPS alone (Fig. 7b). These findings reveal that LPS *E. coli* and *S. minnesota* differentially activate MyD88-dependent pathways.

To uncover potential differences in the recruitment of the adapter protein MyD88 by LPS *E. coli* and *S. minnesota*, we first transfected HEK293 TLR4/MD-2/CD14 cells with Venus-tagged MyD88 and assessed cellular localization of MyD88 by immunofluorescence analysis. In the absence of LPS, MyD88 was located in condensed form in the cytoplasm, as previously demonstrated[51].

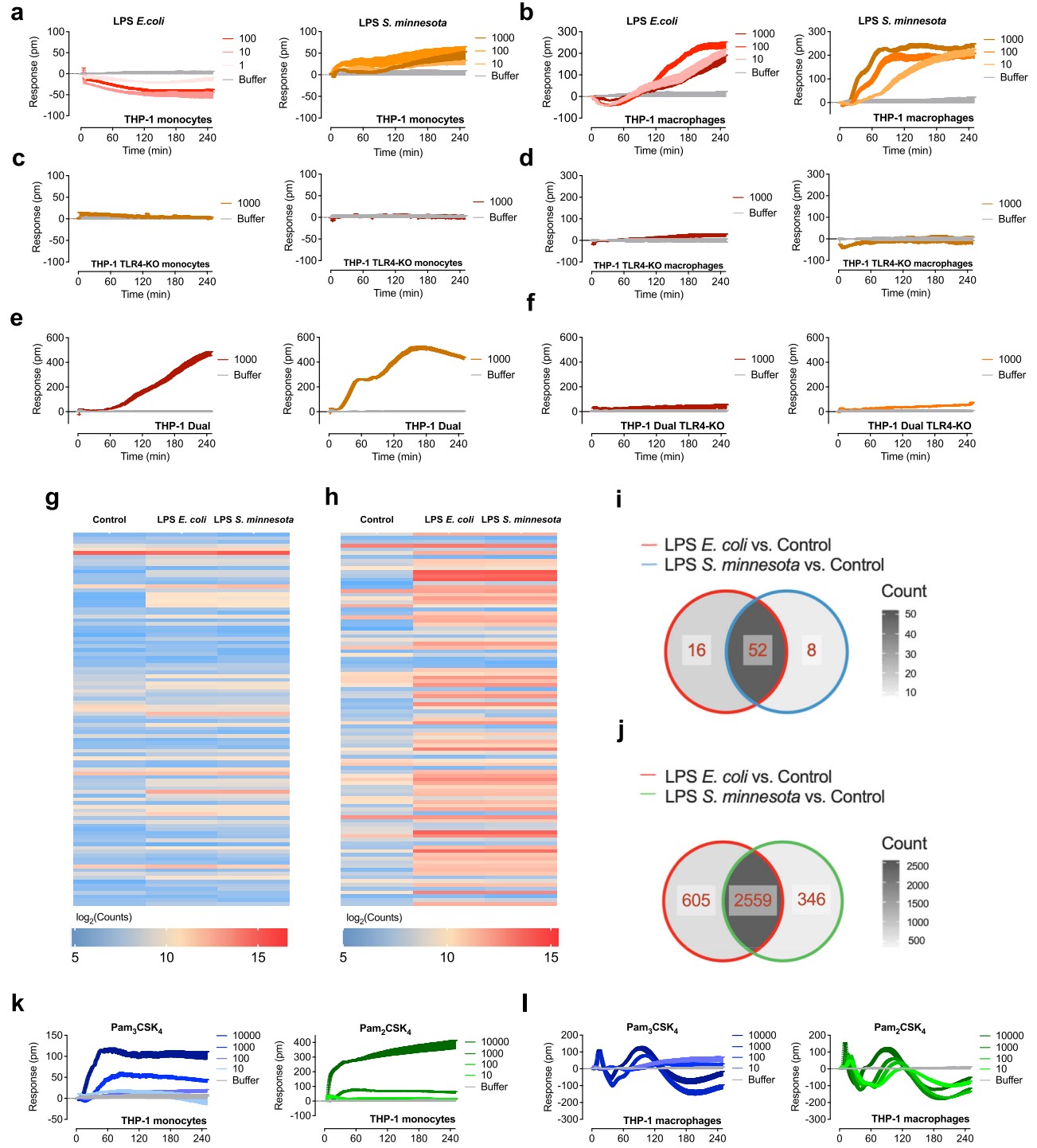

**Fig. 5 | Optical biosensor reveals ligand bias in THP-1 monocytes and macrophages.** THP-1 monocytes (**a**) or macrophages (**b**) were stimulated with increasing concentrations (ng/ml) of LPS *E.coli* and LPS *S. minnesota*. THP-1 KO-TLR4 monocytes (**c**) or macrophages (**d**) were stimulated with increasing concentrations (ng/ml) of LPS *E. coli* and LPS *S. minnesota*. THP1-Dual TLR4/MD-2/CD14 (THP-1 Dual) (**e**) or THP1-Dual KO-TLR4/MD-2/CD14 (THP-1 Dual TLR4-KO) (**f**) cells stimulated with increasing concentrations (ng/ml) of LPS *E. coli* and LPS *S. minnesota*. Heatmap of the top 100 significant up- or downregulated genes identified in HEK293 TLR4/MD-2/CD14 cells (**g**) or THP-1 macrophages (**h**) treated with buffer only (control), LPS *E. coli* or LPS *S. minnesota*, after 3 h incubation. The global transcriptional response induced by the LPS chemotypes showed no significant differences. *n* = 2 biologically independent experiments. Venn diagram for HEK293 TLR4/MD-2/CD14 cells (**i**) and THP-1 macrophages (**j**) indicating the number of significant (FDR < 0.05) differentially expressed genes and the overlap between each set of genes treated with buffer only (control), LPS *E. coli* or LPS *S. minnesota*, after 3 h incubation. **k, l** THP1 monocytes (**k**) or macrophages (**l**) were stimulated with increasing concentrations (ng/ml) Pam$_3$CSK$_4$ or Pam$_2$CSK$_4$. Baseline-corrected DMR recordings are mean + SEM and representative of three biologically independent experiments. Source data are provided as a Source Data file.

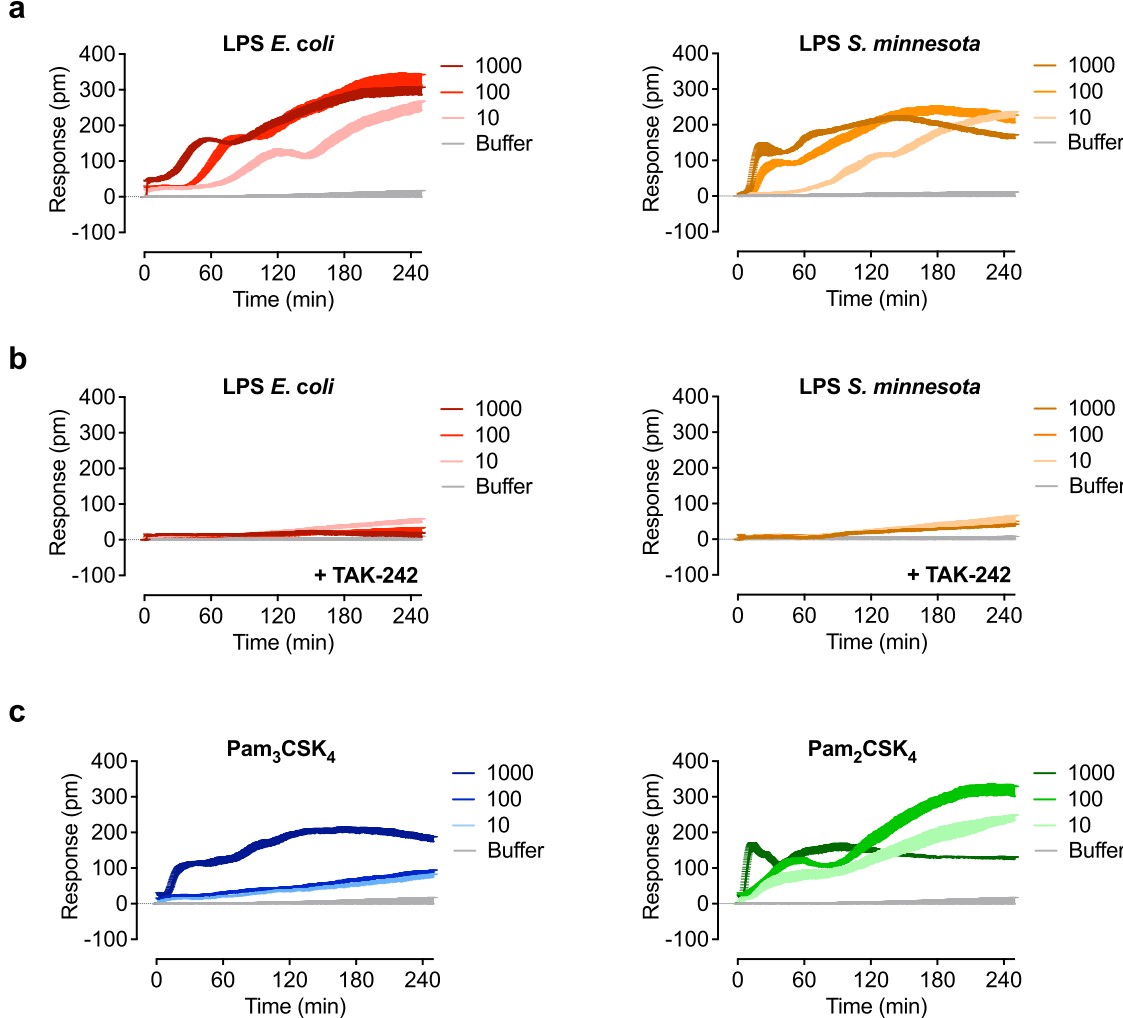

**Fig. 6 | Optical biosensor assay decodes signaling in primary monocytes.**
**a** Primary monocytes isolated from PBMCs were stimulated with the indicated concentrations (ng/ml) of LPS from *E. coli* or *S. minnesota*. **b** Primary monocytes isolated from PBMCs were preincubated with 50 μM of the TLR4 antagonist TAK-242 and stimulated with the indicated concentrations (ng/ml) of LPS from *E. coli* or *S. minnesota*. **c** Primary monocytes isolated from PBMCs stimulated with the indicated concentrations (ng/ml) of Pam$_3$CSK$_4$ and Pam$_2$CSK$_4$. Baseline-corrected DMR recordings are mean + SEM and representative of three biologically independent experiments and donors. Source data are provided as a Source Data file.

However, after LPS stimulation, MyD88 redistributed throughout the cell (Supplementary Fig. 13a). To determine whether this condensation, which does not reflect the behavior of MyD88 under physiological conditions, was due to MyD88 overexpression, we generated HEK293 MyD88-KO cells, which lack functional TLR4 activity (Supplementary Fig. 13b–d). HEK293 MyD88-KO cells transfected with TLR4/MD-2/CD14 and endogenous amount of MyD88-Venus (Supplementary Fig. 13e) showed a similar condensation pattern of MyD88 in the cytoplasm (Fig. 7c). After stimulation with LPS *E. coli*, MyD88-Venus readily dispersed from the condensed structures and formed puncta within 5 min. In contrast, LPS *S. minnesota* induced a delayed formation of MyD88-Venus puncta (Fig. 7c, white arrows). By 45 min, the formation of puncta appeared complete for both LPS chemotypes. MyD88-transfected HEK293 MyD88-KO lacking TLR4 showed MyD88 in condensed form also in the presence of LPS (Supplementary Fig. 13f). The differences in the kinetics of MyD88 assembly induced by LPS *E. coli* and *S. minnesota* might correspond to the signal traces observed by the optical biosensor (Fig. 7d).

**Optical biosensor assay decodes signaling of endosomal TLRs**
In contrast to other TLRs that are typically located on the cell surface and identify bacterial danger signals, a subset of TLRs, including TLR3

and TLR8, are located within the cell endosomes (Fig. 8a). We investigated whether endosomal TLR signaling could also be detected using optical biosensor technology. HEK293 cells express endogenous TLR3[52] and the selective ligand poly(A:U)[53] induced a concentration-dependent negative response (Fig. 8b). High concentrations (100 and 250 μg/ml) did not return to baseline during the recording period. To further characterize the concentration-dependent response of poly(A:U), concentration-effect curves calculated from the area under the curve were generated and an EC$_{50}$ value of 45 μg/ml was determined (Fig. 8c). In the absence of selective TLR3 inhibitors, we used the antimalarial drug and endosomal TLR inhibitor chloroquine[54] to confirm the poly(A:U)-induced signal. Chloroquine alone showed no significant effect (Supplementary Fig. 14a) and clearly inhibited the poly(A:U)-induced signal (Fig. 8d).

TLR8 senses viral and bacterial uridine-rich single-stranded RNA within the endosomal compartment[55–58] (Fig. 8a). The potent and selective TLR8 agonist TL8-506 triggered also a negative DMR signal in TLR8-transfected HEK293 cells with a plateau at 25 min (Fig. 8e). A concentration-effect curve calculated from the area under the curve of DMR traces showed an EC$_{50}$ value of 0.2 μM (Fig. 8f) which is in line with the EC$_{50}$ of 0.6 μM generated in a secreted embryonic alkaline phosphatase reporter assay[59]. CU-CPT9a, a highly potent and

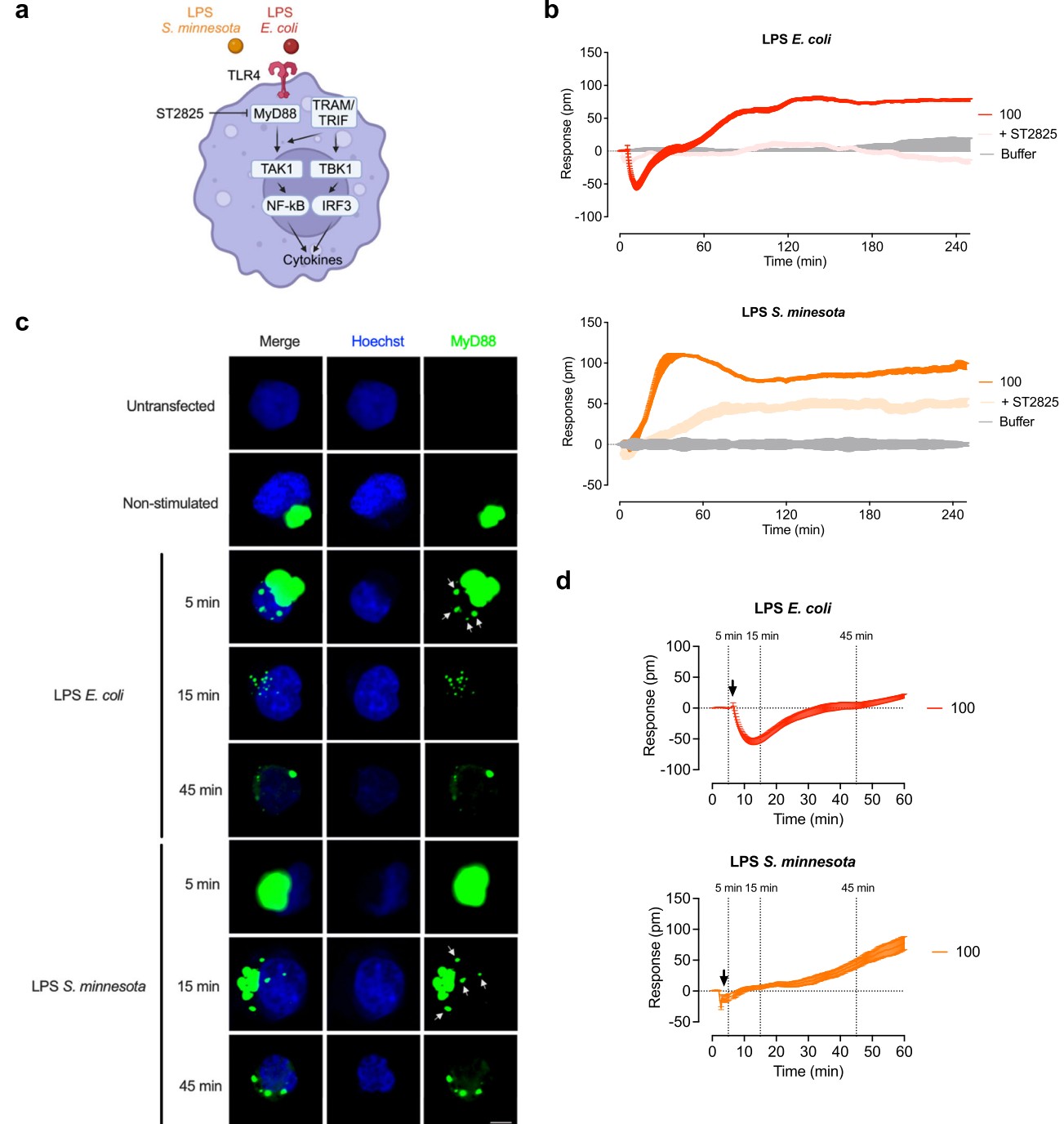

**Fig. 7 | MyD88 inhibition confirmed differences in signaling induced by different LPS chemotypes. a** Schematic representation of TLR4 signaling, ligands used and contact point of the MyD88 inhibitor ST2825. HEK293 TLR4/MD-2/CD14 reporter cells were stimulated with LPS from *E. coli* (100 ng/ml) (**b**) or *S. minnesota* (100 ng/ml) or preincubated with 10 μM of the MyD88 inhibitor ST2825. **c** Immunofluorescence microscopy for localization experiments of MyD88 (green) in transfected HEK293 KO-MyD88 cells before and after stimulation with LPS *E. coli* and LPS *S. minnesota* (100 ng/ml) for 5 min, 15 min or 45 min. Cells are transfected with 500 ng MyD88-Venus construct and counterstained with the nuclear probe Hoechst (blue). Scale bars, 5 μm. Images are representative of three biologically independent experiments. **d** HEK293 TLR4/MD-2/CD14 reporter cells were stimulated with LPS from *E. coli* (100 ng/ml) or *S. minnesota* (100 ng/ml). Baseline-corrected DMR recordings are mean + SEM and representative of three biologically independent experiments. Source data are provided as a Source Data file. (**a**) was created in BioRender. Weindl, G. (2024) BioRender.com/x63i940.

selective TLR8 inhibitor[60,61] completely inhibited the TL8-506-induced signals (Fig. 8g) without producing a DMR shift (Supplementary Fig. 14b) indicating that the signal is TLR8 receptor-specific. No signals were detected in the presence of the TLR8 agonist in HaCaT or control HEK293 cells lacking TLR8 (Supplementary Fig. 14c, d).

## Discussion

Despite many efforts to decipher the precise signaling mechanisms of TLRs, large gaps remain in our understanding of TLR function and signaling[5]. The development of TLR-based therapies has been partially hampered by incomplete knowledge of signaling complexities as exemplified by the failure of TLR4 antagonists in sepsis[62,63]. Receptor

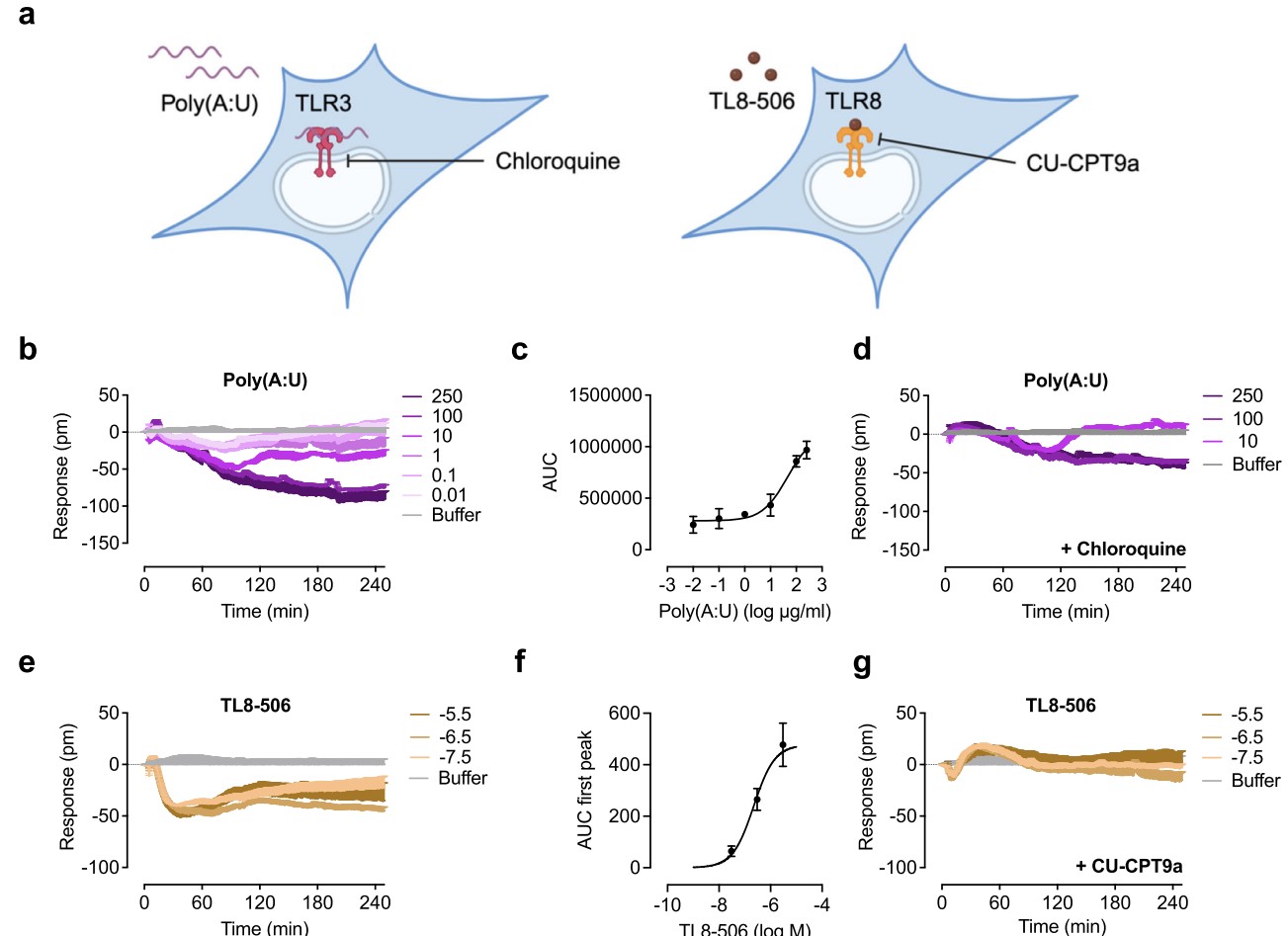

**Fig. 8 | Optical biosensor decodes signaling of endosomal TLRs. a** Schematic representation of TLR3 and TLR8 signaling, ligands used and contact point of the inhibitors chloroquine and the antagonist CU-CPT9a. **b** HEK293 control reporter cells were stimulated with the indicated concentrations (μg/ml) of the TLR3 agonist poly(A:U). Sigmoidal concentration effect curves resulting from DMR traces of $n$ biologically independent experiments (**c**: $n = 4$ biological replicates except log 1, log −1 $n = 3$ (Mean ± SEM)). Concentration-effect curves of DMR data were generated by the area under the curve (AUC). **d** HEK293 control reporter cells were preincubated with 15 μM chloroquine and stimulated with the indicated concentrations (μg/ml) of poly(A:U). **e** HEK293 TLR8 reporter cells were stimulated with the indicated concentrations of the TLR8 agonist TL8-506. **f** Sigmoidal concentration effect curves resulting from DMR traces of $n = 5$ biologically independent experiments (Mean ± SEM). Concentration-effect curves of DMR data were generated by the area under the curve (AUC). **g** HEK293 TLR8 reporter cells were preincubated with 1 μM CU-CPT9a and stimulated with the indicated concentrations of TL8-506. Baseline-corrected DMR recordings are mean + SEM and representative of three biologically independent experiments. Source data are provided as a Source Data file. (**a**) was created in BioRender. Weindl, G. (2024) BioRender.com/g76e041.

signaling, including TLRs, is often promiscuous and triggers several intracellular pathways after receptor activation by endogenous or exogenous ligands. Label-free DMR capturing morphological rearrangement enables real-time detection of integrated cellular responses in living cells and is a powerful tool for studying receptor activation and pathway signaling[6,39,42,64]. We are now able to visualize signals directly after TLR activation and potentially uncover previously unrecognized mechanisms in TLR pharmacology. Optical biosensor technology identifies distinct spatiotemporal dynamics induced by ligands for cell surface and endosomal TLRs and cell type-dependent signaling even in primary cells. This technology also provides $EC_{50}$ and $E_{max}$ values, facilitating the determination of drug potency and efficacy.

Dimerization of TLRs triggers the assembly of supramolecular organizing centers in the cytosol, e.g., the myddosome and triffosome, leading to inflammatory and adaptive immune responses[5,65]. Although signaling cascades have been extensively studied, it is still unclear whether different TLR ligands targeting the same receptor show bias towards intracellular signaling pathways. Ligands can bias protein conformation through selective affinity, which differentially activate

certain receptor signaling responses compared with others[14]. Ligand bias has been mainly studied for GPCRs and only very limited evidence is available for TLR4 and TLR9 ligands[66–68]. TLR4 activates MyD88-dependent and TRIF-dependent signaling pathways[47,48]. The ability to selectively target a single signaling pathway offers tools for fine-tuning of receptor signaling, and perhaps even receptor conformations. We demonstrate that different LPS chemotypes induce distinct DMR responses, suggesting the possibility of biased signaling downstream of TLR4[69]. Our results are unequivocally in line with structure-activity relationship studies that show that structural differences in lipid A can result in varying degrees of inflammation. Lipid A from *E. coli* maximally stimulates the inflammatory response whereas lipid A from *S. minnesota* induces a lower inflammatory response through NF-κB and a greater stimulatory effect on IRF3 signaling[70–72]. These findings indicate that structural differences not only influence potencies but also result in biased agonism[16]. Unlike GPCR signaling, it might be challenging to precisely pinpoint a signal trace to a specific TLR downstream pathway, we were able to uncover kinetic differences of LPS chemotypes in MyD88 activation. The dynamics of cellular

responses including activation of NF-κB, a major transcription factor involved in TLR signaling, is cell type dependent[73]. The different DMR signals obtained in epithelial, epidermal and immune cells induced by TLR4 and TLR2 ligands, respectively, underline the unique spatio-temporal dynamics. While it is widely accepted that TLR4 and TLR2 recognize different cell wall components during bacterial infection and share the same downstream NF-κB signal processing apparatus[74,75], our findings reveal that inducing the same signal partner does not necessarily result in the same signaling pathway dynamics or kinetics. When stimulated independently by specific TLR4 and TLR2 agonists, distinct dynamic NF-κB profiles appeared in single cells[76]. These findings are consistent with signal traces detected with optical biosensor assays, which also allow for the characterization of NF-κB dynamics induced by TLR ligands.

Based on the results obtained with cytoskeleton inhibitors, we concluded that the LPS signal detected by the optical biosensor originated from changes in actin and microtubule rearrangement downstream of TLR activation, which is consistent with previous reports[77,78]. Thus, the optical biosensor technology used here confirmed the link between TLR activation and the reorganization of the actin cytoskeleton and microtubules.

Furthermore, optical biosensor technology uncovers differential TLR2 signaling between monocytes and macrophages[79] during a single experiment. These results suggest that activation of TLR heterodimers by the different agonists induce different signaling cascades/kinetics in monocytes and macrophages and provide evidence for ligand- and system-specific bias which has been reported for GPCRs[80]. Real-time analysis also provides opportunities to visualize specific signaling patterns in cells that are equipped with different signaling molecules.

The optical biosensor assay is sufficiently sensitive to detect ligand-specific optical traces, as demonstrated by the delayed activation kinetics of the synthetic lipopeptide $Pam_3CSK_4$ at TLR2/1 compared to $Pam_2CSK_4$ at TLR2/6[81]. These findings confirm the possibility of ligand-induced differences between the heterodimers. Crystal structures have revealed that the activated TLR2/TLR1 and TLR2/TLR6 heterodimers, when bound to triacetylated (e.g., $Pam_3CSK_4$) and diacetylated (e.g., $Pam_2CSK_4$) lipopeptides, respectively, have distinct binding pockets in their extracellular ectodomains[82,83]. While optical biosensor technology does not determine whether the observed differences stem from variations in ligand binding affinities or differences in signaling pathway activities, it clearly demonstrates that the two ligands behave differently. $Pam_3CSK_4$ and $Pam_2CSK_4$ lead to distinct signatures after TLR2/1 activation suggesting that TLR2 heterodimers induce ligand-dependent biased signaling. This finding has remained elusive using conventional end-point assays but is uncovered by optical biosensor technology.

TLR3, TLR7, TLR8, TLR9 are expressed intracellularly within the endoplasmic reticulum, endosomes and lysosomes[84]. A unique feature of TLR3 is the ability to signal independently of the adaptor protein MyD88[85]. Activation of the receptor leads to the recruitment of the adaptor TRIF which associates subsequently with TRAF3 and TRAF6. Further steps activate both NF-κB, MAPKs[86,87] and IRF3[88]. Given the evidence that chloroquine reduces TLR3/IRF3/IFN-β signaling[89] the residual signal after incubation with chloroquine reflects a pathway that is still active. Different signal traces following TLR3 and TLR8 activation may also provide evidence for the involvement of different adaptor proteins, e.g., MyD88. It is likely that biased signaling is initiated at endosomal TLRs by nucleic acids of varying origins to fine-tune innate immune responses.

Biosensor technology can uncover the complexity of signaling pathways, revealing optical signatures influenced by numerous mediators. However, it is important to recognize that label-free responses are often considered a 'black box' technology. Significant preliminary work with conventional assays and selective ligands is still required before optical biosensors can be fully used as a label-free method, allowing its advantages to be realised.

When combined with pharmacological pathway inhibitors or advanced genome editing tools such as CRISPR/Cas9 technology, biosensor technology can help identify specific proteins involved in TLR signaling, as reflected by DMR signals, thus complementing traditional experimental assays. Our study highlights the potential of this methodology to discover activation mechanisms in real time, within living cells, and without the need for tags. This could provide an additional dimension for understanding TLR biology and pharmacology.

Our approach is also an option to investigate other pattern recognition receptors such as NOD-like receptors and RIG-I-like receptors and TLR agonists or antagonists on patient tissues, reflecting disease-relevant conditions[90,91]. By using optical biosensor technology differences in signaling pathway activation in the tissue removed from the patient could be immediately recognized. Together with the possibility to monitor the cellular responses in real-time this method could give indications of illnesses or changes and thus further deepens the understanding of dysregulated TLR signaling.

Real-time analysis of receptor dynamics provides a comprehensive understanding of events that occur earlier than what is typically observed with assays that report an effect at a specific endpoint or signaling step at several fixed time points. Assays based on optical biosensor technology allow for the detection of signaling events that have been previously overlooked and readily identify and characterize ligands that act on the same receptor but differentially activate downstream pathways. Our findings offer an unprecedented approach and warrant further investigation of the dynamics of TLR signaling to foster the development and characterization of TLR modulators.

## Methods

### Ethics statement
The studies with human blood were approved by the ethics committee of the University Clinic Bonn (315/22) and written informed consent was obtained from all healthy donors.

### Chemical compounds
The TLR ligands $Pam_3CSK_4$ and $Pam_2CSK_4$ (synthetic triacylated and diacylated lipoproteins), lipopolysaccharide from *E. coli* O111:B4 (LPS *E. coli*), *S. minnesota* R565 (LPS *S. minnesota*) and *R. sphaeroides* ultrapure, TL8-506, CU-CPT9a as well as poly(A:U) were purchased from InvivoGen (Toulouse, France). TAK-242 was from Bio-Techne GmbH (Wiesbaden, Germany). Acetylcholine iodide, iperoxo and atropine sulfate, epinephrine, forskolin, chloroquine-diphosphate, the TLR2 antagonist CU-CPT22, as well as the actin inhibitor cytochalasin B and the tubulin inhibitor nocodazole were purchased from Sigma Aldrich Chemie (Steinheim, Germany). The TLR2 antagonist C29 was obtained from ChemDiv (San Diego, USA). The inhibitor ST2825 (MyD88) was obtained from MedChemExpress (New Jersey, USA). The actin inhibitor latrunculin A was obtained from Biomol GmbH (Hamburg, Germany). The chemical structures are shown in Supplementary Fig. 15.

### Cloning of MyD88-Venus construct
MyD88 was synthetized and inserted into the HindIII and XhoL restriction sites of vector plasmid pcDNA3.1 by Genecust (Boynes, France). Venus-Ggamma-pcDNA3.1 was a gift from Kirill Martemyanov. cDNA encoding human MyD88, including a N-terminally HindIII restriction site and a C-terminally GGATCC linker, was amplified by PCR (primers: GTCAGCAAGCTTATGGCCGCTGGCG-GACCTGG, GCTCCTCGCCCTTGCTCACCATG GATCCACCTCCGGG-CAGGCTCAGGGCTTTGGC). XhoI was inserted C-terminally into the

cDNA of Venus using amplification by PCR (primers: ATGGTGAG-CAAGGGCGAGGAGC, GCTGACCTCGAGTTACTTGTACAGCTCGT CCA TGCCGAG). The plasmid expressing C-terminally Venus-tagged MyD88 (MyD88-Venus) was generated by overlap-extension PCR of the amplified MyD88 and Venus coding sequences (primers: GTCAGCAAGCTTATGGCCGCTGGCGGACCTGG, GCTGACCTCGAGT-TACTTGTACAGCTCGTCCATGCCGAG) and subcloning into the Hin-dII and XhoI sites of pcDNA3.1.

## Cell culture and primary cells

HEK-Blue hTLR2-TLR1 (#hkb-htlr21), hTLR2-TLR6 (#hkb-htlr26), hTLR2KO-TLR1/6 (#hkb-htlr2k16), hTLR4 (#hkb-htlr4), hTLR8 (#hkb-htlr8) and Null2 cells (#hkb-null2) (InvivoGen, Toulouse, France) were cultured in DMEM medium supplemented with 10% fetal bovine serum (FBS), penicillin (100 U/ml), streptomycin (100 µg/ml), L-glutamine (2 mM), normocin (100 µg/ml) and selective antibiotics (selection: hTLR2/1, hTLR2/6, hTLR4; zeocin (100 µg/ml): hTLR8, Null2; blasticidin (30 µg/ml): hTLR8). HEK293 cells (#300192, CLS Cell Lines Service, Eppelheim, Germany) were grown in DMEM high glucose supplemented with 10% FCS, l-glutamine (2 mM), 100 U/ml penicillin, and 100 µg/ml streptomycin and HaCaT cells (#300493, CLS Cell Lines Service) were cultured in RPMI 1640 supplemented with 10% FCS, l-glutamine (2 mM), 100 U/ml penicillin, and 100 µg/ml streptomycin. THP-1 cells (passage 4–25) (#ACC 16, DSMZ, Braunschweig, Germany) were cultured in RPMI 1640 supplemented with 10% FCS, l-glutamine (2 mM), 100 U/ml penicillin, and 100 µg/ml streptomycin as described previously[92,93]. THP1-Dual TLR4/MD-2/CD14 (#thpd-mctlr4) and THP1-Dual KO-TLR4/MD-2/CD14 (#thpd-mckotlr4) cells (InvivoGen, Toulouse, France) were cultured in RPMI-1640 medium supplemented with 10% fetal bovine serum (FBS), penicillin (100 U/ml), streptomycin (100 µg/ml), L-glutamine (2 mM), HEPES (25 mM), normocin (100 µg/ml) and the selective antibiotics blasticidin (10 µg/ml) and zeocin (100 µg/ml) following the manufacturer's instructions. THP-1 TLR4-KO cells were grown in RPMI-1640 medium supplemented with 10% fetal bovine serum (FBS), penicillin (100 µ/ml), streptomycin (100 µg/ml), L-glutamine (2 mM), non-essential amino acids and sodium pyruvate (1 mM). Cells were kept at 37 °C in a humidified atmosphere with 5% $CO_2$. HEK-Blue Null2 and hTLR2KO-TLR1/6 cells were used as control as previously described[34,94,95].

PBMCs were isolated from buffy-coat donations (Institute of Experimental Haematology and Transfusion Medicine, University Clinic Bonn) by density gradient centrifugation as described previously[96] using Biocoll separation media (Bio&Sell, Nuremberg, Germany). PBMCs were washed three times with PBS containing EDTA. After the third washing step, $5 \times 10^6$ (24-well) or 70.000 (384-well) PBMCs were seeded per well to isolate the contained primary monocytes due to plastic adhesion. After 1 h incubation at 37 °C in a 5% $CO_2$ incubator the remaining suspension cells were removed from the adherent primary monocytes by washing the cells three times with PBS. Before stimulation, the cells were incubated in media with 10% FCS and antibiotics (penicillin/streptomycin) for 24 h at 37 °C in a 5% $CO_2$ incubator.

## Generation of knockout cells with CRISPR-Cas9

The HEK293 MyD88-KO cell line was created using an eSpCas9-2A-GFP plasmid (GenScript) containing the gRNA sequence 5′-CGAC-GACGTGCTGCTGGAGC-3′. As a negative control, wild-type (WT) HEK293 cells were transfected with the corresponding empty eSpCas9-2A-GFP plasmid. Cells were transfected with the plasmid by using PEI (Polysciences) and were cultured at 37 °C in a humidified atmosphere with 5% $CO_2$. 48 h after transfection single cells were seeded by limiting dilution. Single cell colonies were subsequently screened by western blot analysis and verified by functional assays.

THP-1 TLR4-KO cell lines were generated using an EF1a-Cas9-U6-sgRNA-2A-EGFP expression plasmid encoding the gRNA sequence 5′-CCTGCGTGAGACCAGAAAGC-3′. Cells were transfected

using the Neon Transfection System (Thermo Fisher Scientific) at 1250 V, 50 ms, 1 pulse. Cells were sorted for EGFP expression and single cells were seeded by limiting dilution. Single-cell colonies were subsequently screened by western blot analysis and verified by Sanger sequencing and functional assays. Genomic DNA surrounding the CRISPR targeting site was amplified by PCR (primers: TLR4: 5′-GGTCTGCAGGCGTTTTCTTC-3′, 5′-CATGCCCCTGTTAGCACTCA-3′) and sequenced by Sanger sequencing (primer: TLR4: 5′-GGTCTGCAGGCGTTTTCTTC-3′).

## Dynamic mass redistribution label-free assay

DMR assays were performed using the EPIC system (Corning) according to established protocols[39,42,97]. Briefly, the day before the assay, cells were seeded at 20,000 cells/well (HEK293 cells), 10,000 cells/well (HaCaT cells) into either an Epic 384-well uncoated or fibronectin-coated glass microplate (Corning, New York, NY, USA) and cultured for 24 h to obtain confluent monolayers. For generation of macrophages, THP-1 and THP-1 TLR4-KO monocytes were seeded (20,000 cells/well) in medium containing 25 ng/ml phorbol 12-myristate 13-acetate (PMA, Sigma-Aldrich) for 48 h and afterwards rested for 24 h. At the day of the assay, cells were washed twice with 50 µl assay buffer (Hank's Balanced Salt Solution (HBSS)) containing 20 mM HEPES (pH 7.0) to remove the cell culture media and centrifuged for 10 s to ensure that no drops adhered to the sides of the well, and that cells were in contact with the bottom (biosensor). The final volume in each well was 30 µl. In the case of THP-1 monocytes, THP-1 TLR4-KO monocytes, THP1-Dual TLR4/MD-2/CD14 and THP1-Dual KO-TLR4/MD-2/CD14 cells, 40,000 cells were centrifuged, resuspended and seeded as suspension cells in assay buffer on the day the experiment was performed. Epic microplates were incubated post cell-seeding for 1.5 h in the EPIC instrument at 37 °C. Serial compound dilutions were made in the same assay buffer. For DMR measurements the Epic biosensor (Corning) was used. After reading baseline, compounds were added using a semiautomatic liquid handler Selma (Analytik Jena AG, Jena, DE). The addition of 10 µl compounds were carried out in a volume of 30 µl/well. DMR signals were recorded for 250 min, and data analyzed and exported with the Epic Analyzer Software (Corning). All DMR signals were baseline corrected. Antagonists or inhibitors were incubated 2 h before agonist injection. Compound responses represented in traces were described as picometer (pm) shifts over time (min) following baseline normalization. All the experiments were carried out at 37 °C. For a detailed description of the methods see ref. 42. DMR experiments were carried out in triplicates or quadruplicates.

## ELISA

Cell culture supernatants were collected and analyzed for IL-8 release using a commercially available ELISA kit (Thermo Fisher Scientific).

## Reporter assays

THP-1 Dual cells were seeded in 96-well plates at a density of $1 \times 10^5$ cells per well and were stimulated immediately with the specified substance. After 24 h incubation in an $CO_2$ incubator, NFκB activity was measured using a commercial QUANTI-Blue solution (InvivoGen, Toulouse, France), and ISRE activity using QUANTI-Luc solution (InvivoGen, Toulouse, France), following the manufacturer's instructions.

For Luciferase assays a total of $4 \times 10^4$ HEK293 or HEK293 MyD88-KO cells were seeded per well in poly-L-lysine (Poly-L-lysine hydrobromide, P6282, Sigma-Aldrich) coated 96-well plates in media containing 10% FCS and antibiotics (penicillin/streptomycin). After 24 h incubation at 37 °C in a 5% $CO_2$ incubator, the media was changed to media with 10% FCS without antibiotics. Next, cells were transiently transfected with or without 50 ng of the human TLR4 (hTLR4, a gift from Ruslan Medzhitov, Addgene plasmid # 13086; http://n2t.net/addgene:13086; RRID:Addgene_13086), 10 ng of CD14 (pcDNA3.1-

hDC14, a gift from Douglas Golenbock, Addgene plasmid # 13645; http://n2t.net/addgene:13645; RRID:Addgene_13645), 10 ng of MD-2 expression vector (pFlag-CMV1-hMD2, a gift from Douglas Golenbock, Addgene plasmid # 13028; http://n2t.net/addgene:13028; RRI-D:Addgene_13028), 15 ng of the endothelial leukocyte adhesion molecule (ELAM) firefly luciferase reporter vector (pGL3-ELAM-luc, a gift from Douglas Golenbock, Addgene plasmid # 13029; http://n2t.net/addgene:13029; RRID:Addgene_13029), 15 ng Renilla luciferase control vector or empty pcDNA3.1 vector. After transfection, cells were incubated for 24 h at 37 °C in a 5% $CO_2$ incubator. For stimulation, cells were incubated with LPS *E. coli* or LPS *S. minnesota* (100 ng/ml). After 6 h cells were lysed in a passive lysis buffer and firefly luciferase and Renilla luciferase activity were measured using the Dual-Glo luciferase Assay System (E2940, Promega, Madison, USA). For normalization Renilla luciferase was used. The data obtained with firefly were divided by Renilla data. To subtract the basal receptor activity and determine the relative luciferase activity (RLU), medium values were subtracted and divided by the medium values without firefly luciferase.

## RNA isolation and qPCR

Total RNA isolation was performed using innuPREP RNA Mini Kit 2.0 (845-KS-2040050, Analytik Jena, Jena, Germany) according to the manufacturer's protocol. cDNA was synthesized (iScript cDNA Synthesis Kit, 1708891, Bio-Rad, Feldkirchen, Germany) and quantitative real-time RT-PCR (qRT-PCR) was performed as previously described[98] using SYBR Green Supermix (Bio-Rad, 1725120). The following program was used: 95 °C for 5 min, followed by 45 cycles of 95 °C for 10 s, 60 °C for 10 s and 72 °C for 10 s. Primers (synthesized by Eurofins Genomics) with the following sequences were used: GAPDH: 5′-CTCTCTGCTCCTCCTGTTCGAC-3′, 5′-TGAGCGATGTGGCTCGGCT-3′, CXCL8: 5′-CAAGAGCCAGGAAGAAACCA-3′, 5′-GTCCACTCTCAAT-CACTCTCAG-3′. Primer sequences for GAPDH and CXCL8 were published previously[99,100]. Gene expression was quantified using the ΔΔCt method and normalized to the housekeeping gene GAPDH.

## LDH assay

LDH assay was performed according to the manufacturer's instructions (CyQUANT LDH Cytotoxicity Assay, Thermo Fisher Scientific). The percentage of LDH release was calculated compared to 100% cell lysis control.

## MTT assay

MTT test was used for determine effects on cell viability in HEK-Blue cells as described before[101]. Cells ($4 \times 10^4$ cells per well) were grown in growth medium overnight (96-well plates). The indicated agents were added and 25 μl MTT solution (final concentration of 0.5 mg/ml) for the last 4 h. After removing supernatants DMSO was added to dissolve the formazan crystals. The optical density was determined at 560 nm on a Mithras LB 940 reader (Berthold Technologies, Germany). Cell viability of the non-stimulated cells was defined as 100%. DMSO (30%, v/v) served as positive control.

## Western blotting

Proteins were extracted from cultured cells following lysis in ice-cold RIPA buffer containing a protease-phosphatase inhibitor cocktail (Cell Signaling). Protein quantification was performed using a bicinchoninic acid assay (Pierce BCA protein assay, Thermo Scientific, Darmstadt, Germany) according to the manufacturer's instructions. Proteins were separated on 10% TGX Stain-Free polyacrylamide gels (BioRad, Hercules, USA) and blotted to a microporous polyvinylidene fluoride membrane (Merck, Darmstadt, Germany). Membranes were incubated with primary antibodies over night at 4 °C and horseradish peroxidase (HRP)-conjugated secondary antibodies for 1 h at room temperature and proteins were visualized with ECL substrate (BioRad, Hercules, USA). The following antibodies were used: rabbit anti-human CD14 (1:1000, Cell Signaling, D7A2T, #56082), rabbit anti-human MyD88 (1:1000, Cell Signaling, D80F5, #4283), rabbit anti-human TRIF (1:1000, Cell Signaling, #4596), mouse anti-human TICAM-2/TRAM (1:500, Bio Legend, #659402), rabbit anti-human TLR4 (1:1000, Cell Signaling, E5D8T, #38519), mouse anti-human β-actin (1:1000, LI-COR Biosciences, #926-42212), goat anti-rabbit HRP-linked (1:3000, BIO-RAD, #1706515), goat anti-mouse IRDye 680RD-linked (1:15000, LI-COR Biosciences, #926-68070), horse anti-mouse HRP-linked (1:2000, Cell Signaling, #7076), goat anti-mouse HRP-linked (1:2000, Cell Signaling, #7074). Uncropped western blots are provided in Supplementary Fig. 16.

## Immunofluorescence

HEK-Blue hTLR4 ($0.85 \times 10^6$ cells per well, Invivogen, Toulouse, France) or HEK293 MyD88-KO ($0.7 \times 10^6$ cells per well) cells were seeded in 6-well plates in media containing 10% FCS and antibiotics (HEK-Blue hTLR4: penicillin/streptomycin/normocin/selection, HEK293 MyD88 KO: penicillin/streptomycin). After 24 h incubation at 37 °C in a 5% $CO_2$ incubator, the media was changed to media with 5% FCS without antibiotics. Subsequently, the cells were transiently transfected with 200 ng (HEK-Blue hTLR4) or 500 ng (HEK293 MyD88-KO) MyD88-Venus construct using PEI Max (Polysciences) according to manufacturer's protocol. HEK293 MyD88-KO cells were additionally transfected with or without 880 ng TLR4, 180 ng CD14, 180 ng MD-2 or empty vector (pcDNA3.1). After transfection cells were incubated for further 24 h at 37 °C in a 5% $CO_2$ incubator. The transfected cells were seeded in poly-L-lysine (Poly-L-lysine hydrobromide, P6282, Sigma-Aldrich) coated 8-well chamber slides in a total of $7 \times 10^4$ cells per well and were incubated for 48 h at 37 °C in a 5% $CO_2$ incubator. For stimulation the cells were incubated with LPS *E. coli* (100 ng/ml) or LPS *S. minnesota* (100 ng/ml) for 5 min, 15 min or 45 min. After stimulation, the cells were fixed using 4% paraformaldehyde (Carl Roth) for 10 min. Between each staining step the cells were washed extensively using PBS. The cells were permeabilized with 0.2% Triton X-100 (Sigma) for 10 min. The nucleus was stained using Hoechst 33342 (1 μg/ml, Thermo Fisher Scientific, 62249) and the cells were mounted in ProLong Glass Antifade Mountant (Invitrogen, P36982). Images were taken with the confocal laser microscope Nikon Ti-E with A1 confocal scanner and were processed using NIS-Elements Viewer 5.21.

## RNA sequencing and data analysis

Total RNA was harvested according to the manufacturer's instructions (innuPREP RNA Mini Kit 2.0, Analytik Jena). RNA libraries were prepared using QuantSeq 3′-mRNA Library Prep Kit (Lexogen) and RNA sequencing was performed with the NovaSeq 6000 (Illumina).

The nf-core rnaseq pipeline[102] (version 3.10.1) was applied for the preprocessing and the quantification of the reads using default parameters. Here the first step is quality and adapter trimming with Trim-Galore (version 0.6.7). This will be followed by aligning the trimmed reads against human genome (GRCh38) with STAR[103] (version 2.7.9a). The aligned data are then used as input for Salmon[104] (version 1.9.0), which employs pseudoalignment to estimate transcript abundances. The transcript-level quantifications will then be aggregated to obtain gene-level expression estimates.

The statistical analysis was executed in the R environment (version 4.2.0)[105]. Given the notable biological variability observed between the two cell lines, HEK293 and THP-1, distinct analyses were undertaken for each cell line. To ensure the robustness of the results, only genes with a minimum count of 10 in at least two samples were utilized for the inference analysis. The Bioconductor package DESeq2[106,107] was employed for the differential gene expression analysis. Subsequently, the Benjamini-Hochberg method was applied to calculate multiple testing-adjusted p-values (false discovery rate, FDR) for each contrast. For visualization, the packages ggplot2[108] was used to generate volcano plots.

## Statistics and reproducibility

All experimental data were analyzed by using Graph Pad Prism 8.0 (GraphPad Software Inc., San Diego, CA, USA). Representative DMR traces are displayed as means + SEM, and quantified data represent mean ± SEM. No statistical method was used to predetermine sample size. No data were excluded from the analyses. The experiments were not randomized. The investigators were not blinded to allocation during experiments and outcome assessment.

To compare two means, statistical significance was based on a Student's $t$ test with $P < 0.05$. Comparisons of groups were performed using one-way-ANOVA analysis with a Tukey–Kramer post-test, or one-sample $t$ test. Data obtained from DMR experiments were analyzed by a four-parameter-logistic function yielding parameter values for a ligand's potency ($\log EC_{50}$) and maximum effect ($E_{max}$). For data normalization, indicated as relative response (%), top levels of concentration effect curves at 250 min of the data set were set 100% and bottom levels 0%.

## Reporting summary

Further information on research design is available in the Nature Portfolio Reporting Summary linked to this article.

## Data availability

All data generated or analyzed during this study are included in the article and its supplementary information files. The RNA-seq data have been deposited in NCBI's Gene Expression Omnibus[109] and are accessible through GEO Series accession number GSE250546. Source data are provided with this paper.

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

## Acknowledgements

This work was partially funded by the German Research Foundation to G.W. (grant RA 895/16-1). We thank Nicole Merten and Leon Steffens for technical support, and Eicke Latz (Deutsches Rheuma Forschungszentrum Berlin) and Eva Bartok (IHT, University Hospital Bonn) for kindly providing reagents. We thank the Next Generation Sequencing Core Facility of the Medical Faculty at the University of Bonn for providing support and instrumentation funded by the Deutsche Forschungsgemeinschaft (DFG, German Research Foundation). Figures 1a, 1b, 3a, 7a, and 8a, and Supplementary Fig. 1 were created with Biorender.com.

## Author contributions

J.H., F.L., and S.S. performed experiments; J.H., F.L., S.S., and G.W. analyzed the data; E.K. provided reagents; G.W. directed the study; J.H., F.L., E.K., and G.W. wrote the manuscript.

## Funding

## Competing interests

The authors declare no competing interests.
