## [Transparent Peer Review file · Nature Communications]

Label-free biosensor assay decodes the dynamics of Toll-like receptor signaling

Corresponding Author: Professor Günther Weindl

Version 1:

Reviewer comments:

Reviewer #1

(Remarks to the Author)

In this paper the authors have investigated the potential application of a new label-free optical biosensor-based assay, routinely used as a label free system to analyse the responses of GPCRs.

Overall there are a number of problems that need to be addressed by the authors particularly around the experimental models. The authors rely heavily on HEK TLR4-Blue cells which lack some components of the TLR signalling pathways and the experiments in macrophages are not validated with receptor knock outs. The technology is interesting and could be very useful, but with the full cohort of experimental controls and validation the authors hypotheses remain unproven.

Major issues:

1. The technology relies on morphological changes as a signal readout. The authors state “these morphological changes can be observed as long as the signaling event results in a rearrangement of the cytoskeleton”

The technology is based on this principal so where are the validation/control experiments to show that TLR signalling affects the cytoskeletal function and induces morphological changes?

2. “Likewise, fluorescent tags or reporter systems that are prone to non-specific interference have been used to study TLR function and signaling”

This statement is disingenuous because many papers where experiments were performed with tagged constructs are fully controlled by assessing the constructs are signalling competent

3. “Other receptor classes, such as TLRs, also trigger multiple signaling pathways, but their bias signaling capabilities have not yet been established”

This is not true as the concept of biased signalling eg MyD88 vs Trif for TLR4 is well established unless the authors mean something else by this statement?

4. “We reveal unrecognized mechanisms of TLR pathway activation and biased signaling, thus providing a completely new insight into TLR signal transduction.”

Unfortunately this is not supported by the data shown by the applicants at the moment.

5. Why use HEK293 reporter cells, stably transfected to express TLR4 (presumably they express MD2 and CD14 as well otherwise TLR4 cannot signal in response to LPS)? TLR4-HEK-blue cells have a notoriously low level of signalling activity such that they are a poor model for TLR4 signalling although this might be of benefit for the authors’ technique which they claim as being highly sensitive. Data generated in these cells is rarely robust and a much better starting point are TLR4 competent and TLR4-/- macrophages which are also commercially available with robust TLR4 signalling competency

6. “Furthermore, we have shown that heterodimerization of TLR2 with TLR1 or TLR6 leads to clearly distinct optical signals, indicating differential signaling behavior.”

Could these data indicate different binding/affinity or efficacy kinetics to the heterodimers?

7. All the THP1 results need to be validated in cells where either TLR4 or CD14 has been deleted to confirm the different claims relating to these proteins (these cell lines are commercially available)

8. RNAseq data: comparing gene expression data sets between HEKs and THP-1s is not a very good approach. HEK cells are not thought to have Tram therefore these cells are not competent for TLR4-Tram-Trif signalling hence the gene expression profiles will be very different between cell lines. The authors would need to demonstrate their HEK cells express Tram to counter this problem. The limitations of HEKs mean they are not a good system to test the authors hypothesis, it would make more sense just to focus on the THP-1 cell data.

9. It is increasingly clear that MyD88 and Trif signalling pathways do not really operate independently of each other but are, in fact, interlinked. TNFa production in response to TLR4 signalling, for example, requires both MyD88 and Trif for efficient cytokine production (doi: 10.1016/j.celrep.2022.111225). The differences in LPS chemotypes seen by the authors may simply reflect differences in efficacy at TLR4 rather than anything else especially given the absence of Tram in HEK cells. This would need to be fully explored to make the data presented by the authors compelling.

10. “We transfected TLR4-HEK293 cells with Venus-tagged MyD88 and assessed cellular localization of MyD88 by immunofluorescence analysis”

Data from these experiments will not reflect what is normally happening to MyD88 in cells because this is now an over expression system (HEKs have endogenous MyD88) and MyD88 will oligomerise and now form complexes in a diverse array of locations within the cell. These experiments would need to be performed in a MyD88 knock out background and carefully controlled such that the expression levels of tagged MyD88 were similar to those seen endogenously.

Reviewer #2

(Remarks to the Author)

I hereby identify myself as Jon Kagan (Harvard Medical School).

In this manuscript, the authors report the use of an optical biosensor assay to study TLR signaling pathways. A significant dataset is presented, which revealed the utility of this assay to discern cellular responses induced by distinct TLR ligands--in distinct cell types. I commend the authors on their creativity and use of this assay, which may become a standard tool for the field.

I have a general question of the molecular underpinnings of what optical biosensor assays are actually reporting. The authors could be asked to provide such data, but I would be fine if they save such analysis for a future study. My only specific query is listed below.

1. Line 135 describes Mal as a Myd88-independent regulator of TLR signaling. This statement runs counter to many studies and should be reconsidered.

Reviewer #3

(Remarks to the Author)

Holze et al. NatComm:

The manuscript by Holze et al. shows a nice study on Toll-like receptor signaling using a label-free method with the help of dynamic mass redistribution (DMR). The work shows and illustrates the importance of the dynamic and kinetic examination of ligand-induced signaling in the cell. DMR is an innovative and sensitive method for observing signaling processes in the cell that does not require any changes to the cell or the target protein in order to obtain a readout. This work opens up new possibilities for the characterization of the under-researched group of Toll-like receptors and the possible future development of drugs that target TLRs and is therefore significant for the field. The methodology is sound. Even if the work still needs to be improved in some areas, I would like to recommend it for publication in Nature Communications after addressing these issues (major revision). I would like to mention the following points:

- I would like to see a bit more specific background on TLRs in the introduction. How many TLRs are known? Can they be classified? Which ones are covered in this study and why? Is there a link to possible diseases (there is talk of possible future drug development later on)?
- I am missing chemical structures for the TLR ligands used in this study. Here I would suggest a figure to be placed in the Supporting Information. The small molecule modulators should definitely be shown. Are there specific structures for the LPS and PamXCSK4 ligands? Where exactly are the structural differences? What's the chemical difference of LPS smooth (s), rough (r)?
- Page 3, line 73: LPS consists of ... (no comma).
- I am torn with the specification of the DMR readout in seconds. From my point of view, a presentation and discussion in minutes would have been the better choice, as the reader (at least me) finds it difficult to imagine an exact period of time with, for example, 15,000 seconds. In addition, the authors describe their results in seconds, sometimes in minutes and occasionally in hours. This is not conducive to comparability. This should be handled consistently. I would prefer minutes, even if I am aware that this means that a number of figures also have to be adjusted in the x-axis. It should be done consistently in any case. The authors can think about the changeover to minutes (not a major issue).
- I think it's really good that the limitations of the method are discussed at the end (black box readout, need of a ligand toolbox and prefindings from other assays, complementary to conventional assays). In my view, it will take a lot of preliminary work on conventional assays and selective ligands to get to a point where DMR can be used as a label-free method and all the advantages mentioned can be utilized. Perhaps this can be added to the existing section. Perhaps at this point the authors can also better emphasize what contribution the method can make to other assays.
- What techniques have been used to investigate TLR signaling pathways to date?
- Page 34, line 812 (Fig.1): "TAK-242" not "TAK.242".
- Page 5: The authors discuss previous observations of heterodimers. What assay was used and what is novel in this experiment?
- How has dimerization been demonstrated in the literature? Was evidence of dimerization provided in this study?
- Figure 1/2 caption: I assume the dashed lines represent the six time points that were used to generate CRCs. If that's true, then please state this in the respective Figure captions.
- Page 5, line 120: "while Pam2CSK4 mainly activated TLR2/6". It should be mentioned here that Pam2CSK4 also triggers significant activation at TLR2/1. Especially as this is discussed in the next chapter (line 155-157). In addition, the designation in Supp. Fig 2a is wrong or at least very confusing. It says "... Pam2CSK4 (TLR 2/6) and Pam3CSK4 (TLR 2/1)", although the test was performed exactly the other way around as indicated in the brackets. It should also be pointed out in the main text that both Pam ligands were also "cross-tested" at the other dimer target.
- In lines 147-148 (Page 6) it is stated: "We used the well-characterized epidermal HaCaT cell line that responds to TLR2/1 and TLR2/6 ligands, but shows no functional TLR4 signaling[34]." But in line 155 "HaCaT cells lack the expression of

TLR6[34]". How does that work?

- Page 7, Line 162: "... overlapping signals can be excluded". That might be true for C29 but the signal CU-CPT22 has a negative response after reaching the baseline (Supp. Fig. 3b). Accordingly, overlapping signals in Fig. 3c cannot be ruled out in my opinion.
- Page 7, Lines 162-163: „The optical traces triggered by the lipopeptides were independent of TLR4 (Supplementary Fig. 3c).“ I do think that the Pam2CSK4 signal has been influenced. The course is similar, but the curve as a whole experiences a clear depression, even into the negative response range. How can this be explained?
- Page 8, line 184-185: Please do not abbreviate "RNA-seq". At least describe it at first appearance.
- Common question: How can negative peaks be interpreted in DMR readouts compared to positive signals? Is there a direct connection to the mode of action, for example when agonists and inverse agonists are compared?
- Page 9, line 225: The small molecule ST2825 is described to be a "... synthetic analogue of (protein) MyD88 ...". It sounds a bit strange to me how a small molecule structure could be an analogue of a protein. Please explain or be more precise. Otherwise this statement/wording is misleading.
- Page 9, line 228: I think the initial negative peak is also abolished by ST2825, isn't it?
- Figure 6b: Indicate the used concentrations (100 ng/mL) as legend instead of "control".
- Figure 6b caption: Use spaces between numbers and units! Check throughout the manuscript.
- I don't think that "blunted" is the right wording for antagonizing the induced effects. This was used at least two times. Please change!
- Page 13, line 33: I think "that" needs to be deleted (Grammar).
- Positive: A number of reasonable control experiments were chosen and executed.
- Table 1: nH is not described (most likely the Hill slope).
- Table 1/2: I find the specification of the LogEC50 calculated from ng/mL concentrations confusing, not very meaningful and difficult to compare. Are no molecular weights given/known for the compounds so that the concentration could be given in moles?

Version 2:

Reviewer comments:

Reviewer #1

(Remarks to the Author)

The authors have performed a number of new experiments to address the issues raised by this referee. A few issues still remain, but these can be resolved by important additions to the text as follows:

1. "To verify that the LPS E. coli signal detected in suspension HEK293 reporter cells, which are stably transfected to express TLR4, is mediated by TLR4, the inhibitor TAK-242 was used (Fig. 2a)."

Were cells transfected with just TLR4 or TLR4/MD2? Without MD2 LPS cannot signal.

2. Were the experiments with THP-1 cells done in the presence of serum? If so the serum will provide soluble CD14. I don't think CD14 is necessarily important for the MS, but the text must be correct so even if the cells don't appear to express CD14, the serum must be taken into account and a comment added accordingly to the text.

3. Fig 7: The m-venus MyD88 aggregates in HEKs are not present in fluorescently tagged MyD88 in macrophages. This suggests the HEK model is a problem. In the opinion of this reviewer data with the m-Venus MyD88 should be removed from the MS. If the authors think it is critical to have these data in the MS then a sentence needs to be added stating that "whilst aggregated tagged MyD88 is what is seen in HEK cells this does not reflect the behaviour of MyD88 under physiological conditions for example in macrophages"

Reviewer #2

(Remarks to the Author)

The authors have addressed my minor comment. No additional concerns are offered. I am enthusiastic for publication.

Reviewer #3

(Remarks to the Author)

The authors have addressed all my concerns in the revision. I only see one more issue. Supplementary Figure 15 is very blurry. Please provide the figure in a much better resolution. Other than that, in my opinion, the manuscript can be published in the current form.

We thank all the reviewers for their constructive feedback and valuable suggestions on the manuscript. In the revised version, we have addressed the points raised by the referees. We believe these revisions have significantly improved the quality of our work and the manuscript.

REVIEWER COMMENTS

Reviewer #1 (Remarks to the Author):

In this paper the authors have investigated the potential application of a new label-free optical biosensor-based assay, routinely used as a label free system to analyse the responses of GPCRs.

Overall there are a number of problems that need to be addressed by the authors particularly around the experimental models. The authors rely heavily on HEK TLR4-Blue cells which lack some components of the TLR signalling pathways and the experiments in macrophages are not validated with receptor knock outs. The technology is interesting and could be very useful, but with the full cohort of experimental controls and validation the authors hypotheses remain unproven.

Response authors: We appreciate the reviewer's careful reading and constructive feedback. In response to the suggestions, we conducted additional experiments, and the findings have been thoroughly incorporated into the revised manuscript and supplementary information. We sincerely thank the reviewer for the interest and valuable comments on our work.

Major issues:

1. The technology relies on morphological changes as a signal readout. The authors state “these morphological changes can be observed as long as the signaling event results in a rearrangement of the cytoskeleton”

The technology is based on this principal so where are the validation/control experiments to show that TLR signalling affects the cytoskeletal function and induces morphological changes?

Response authors: We fully agree with the point made by the reviewer and have now collected data demonstrating the cytoskeleton dependence of the signaling events. The revised manuscript text includes new data (Fig. 2^{NEW}, Supplementary Fig. 4^{NEW}) and interpretation in the results section:

“To confirm that the changes induced by LPS affect cytoskeletal function and lead to morphological alterations, we preincubated cells with inhibitors of actin or tubulin polymerization. Since inhibiting actin- or tubulin-dependent cytoskeletal restructuring could cause cell detachment, appearing as a loss of mass, the experiment was conducted in suspension mode rather than adherent mode. Sufficient time was allowed for baseline equilibration before adding LPS E. coli (Supplementary Fig. 4a-c). To verify that the LPS E. coli signal detected in suspension HEK293 reporter cells, which are stably transfected to express TLR4, is mediated by TLR4, the inhibitor TAK-242 was used (Fig. 2a). Both actin inhibitors, cytochalasin B²³ and latrunculin A²⁴, as well as the microtubule inhibitor nocodazole²⁵, reduced the LPS E. coli-mediated response in a concentration-dependent manner (Fig. 2b-d). Throughout the detection period, cells remained viable in the presence of the inhibitors (Supplementary Fig. 4d).”

And in the discussion:

“Based on the results obtained with cytoskeleton inhibitors, we concluded that the LPS signal detected by the optical biosensor originated from changes in actin and microtubule rearrangement downstream of TLR activation, which is consistent with previous reports^{76,77}. Thus, the optical biosensor technology used here confirmed the link between TLR activation and the reorganization of the actin cytoskeleton and microtubules.”

2. “Likewise, fluorescent tags or reporter systems that are prone to non-specific interference have been used to study TLR function and signaling”

This statement is disingenuous because many papers where experiments were performed with tagged constructs are fully controlled by assessing the constructs are signalling competent

Response authors: We have rephrased the corresponding statement in the revised manuscript:

“Similarly, investigations frequently use fluorescent tags or reporter systems, which, while well-characterized and invaluable for experimentation, require careful and time-consuming validation. This is essential to prevent potential interference with the natural protein structure, function, or the proper activation of signaling pathways.”

3. “Other receptor classes, such as TLRs, also trigger multiple signaling pathways, but their bias signaling capabilities have not yet been established”

This is not true as the concept of biased signalling eg MyD88 vs Trif for TLR4 is well established unless the authors mean something else by this statement?

Response authors: Thank you for pointing this out. We have rewritten the sentence accordingly:

“The TLR family is also known to activate multiple signaling pathways; however, their bias signaling capabilities are not yet well-defined, and our understanding of TLR ligand bias is less advanced compared to our knowledge of GPCR ligand bias^{13,14}.”

4. “We reveal unrecognized mechanisms of TLR pathway activation and biased signaling, thus providing a completely new insight into TLR signal transduction.”

Unfortunately this is not supported by the data shown by the applicants at the moment.

Response authors: We have rephrased the corresponding sentence to make this clearer:

“Our data provide evidence for distinct mechanisms of TLR pathway activation that occur immediately after receptor engagement, along with potential ligand bias, offering new insights into TLR signal transduction.”

5. Why use HEK293 reporter cells, stably transfected to express TLR4 (presumably they express MD2 and CD14 as well otherwise TLR4 cannot signal in response to LPS)? TLR4-HEK-blue cells have a notoriously low level of signalling activity such that they are a poor model for TLR4 signalling although this might be of benefit for the authors’ technique which they claim as being highly sensitive. Data generated in these cells is rarely robust and a much better starting point are TLR4 competent and TLR4-/- macrophages which are also commercially available with robust TLR4 signalling competency

Response authors: We appreciate the comments and suggestions. We agree that HEK293 cells are a poor model to study TLR signaling. As indicated by the reviewer, we used HEK293 as model system to establish the optical biosensor assay. We have rephrased the relevant results section:

“To assess whether LPS-induced signals can be detected by the optical biosensor, we used HEK293 cells, which endogenously express TRAM and TRIF (Supplementary Fig. 2), and were stably transfected to express TLR4, MD-2 and CD14.”

We have now included data from THP-1 TLR4-KO cells and Dual THP1-Dual MD2-CD14 KO-TLR4 cells. For further details, please refer to comment 7.

6. “Furthermore, we have shown that heterodimerization of TLR2 with TLR1 or TLR6 leads to clearly distinct optical signals, indicating differential signaling behavior.”

Could these data indicate different binding/affinity or efficacy kinetics to the heterodimers?

Response authors: We thank the reviewer for pointing this out. We have now incorporated these considerations into the discussion:

“These findings confirm the possibility of ligand-induced differences between the heterodimers. Crystal structures have revealed that the activated TLR2/TLR1 and TLR2/TLR6 heterodimers, when bound to triacetylated (e.g., Pam₃CSK₄) and diacetylated (e.g., Pam₂CSK₄) lipopeptides, respectively, have distinct binding pockets in their extracellular ectodomains^{81,82}. While optical biosensor technology does not allow to determine whether the observed differences stem from variations in ligand binding affinities or differences in signaling pathway activities, it clearly demonstrates that the two ligands behave differently.”

7. All the THP1 results need to be validated in cells where either TLR4 or CD14 has been deleted to confirm the different claims relating to these proteins (these cell lines are commercially available)

Response authors: As suggested by the reviewer, we have used THP-1 TLR4-KO cells (Fig. 5c,d^{NEW}, Supplementary Fig. 8^{NEW}) and included data from Dual THP1-Dual MD2-CD14 KO-TLR4 cells Fig. 5e,f^{NEW}, Supplementary Fig. 9^{NEW}). To our knowledge, THP-1 CD14-KO cells are not readily available commercially, and no data on them have been published to date. In response to the reviewer’s request, we characterized THP-1 cells for CD14 expression (Supplementary Fig. 7^{NEW}) and used THP-1 Dual MD2-CD14-TLR4 cells. The results section has been revised accordingly:

*“Given that THP-1 monocytes express little⁴⁴ to no CD14 (Supplementary Fig. 7), our findings are in line with these observations. To confirm our results, we generated THP-1 TLR4-KO cells, which were unable to induce cytokine expression in response to LPS *E. coli* but retained the ability to respond to the TLR2/1 ligand Pam₃CSK₄ (Supplementary Fig. 8). No DMR signals were detected after incubation with LPS *E. coli* and LPS *S. minnesota*, confirming that the signals detected in THP-1 cells are mediated by TLR4 (Fig. 5c, d). To further investigate the signals in THP-1 cells and the role of CD14, we used THP1-Dual MD2-CD14-TLR4 cells (hereafter referred to as THP-1 Dual cells). These cells stably express CD14 and we expect them to produce signals similar to those in THP-1 macrophages. LPS *E. coli* and LPS *S. minnesota* induced signals in THP-1 Dual cells nearly identical to those observed in THP-1 macrophages (Fig. 5e), which also express CD14 (Supplementary Fig. 7). No traces could be recorded in THP1-Dual MD2-CD14 KO-TLR4 cells (hereafter referred to as THP-1 Dual TLR4-KO cells) (Fig. 5f) and with LPS *R.**

sphaeroides (Supplementary Fig. 9a, b), whereas TLR2-induced responses remained intact (Supplementary Fig. 9c, d). The obtained DMR signals corresponded with NF- κ B (Supplementary Fig. 9e, g) and IRF activity (Supplementary Fig. 9f, h).”

8. RNAseq data: comparing gene expression data sets between HEKs and THP-1s is not a very good approach. HEK cells are not thought to have Tram therefore these cells are not competent for TLR4-Tram-Trif signalling hence the gene expression profiles will be very different between cell lines. The authors would need to demonstrate their HEK cells express Tram to counter this problem. The limitations of HEKs mean they are not a good system to test the authors hypothesis, it would make more sense just to focus on the THP-1 cell data.

Response authors: We agree that HEK293 cells are not ideal to study TLR signaling; however, we believe these cells are a valuable control cell model to establish the optical biosensor assay. We used HEK293 cells side-by-side with THP-1 cells throughout the manuscript to compare the biosensor results with established assays including RNA sequencing.

As suggested by the reviewer, we characterized HEK293 cells for TRAM and TRIF expression and found endogenous expression of both adapter proteins in HEK293 TLR4 and Null2 cells (Supplemental Fig. 2^{NEW}).

9. It is increasingly clear that MyD88 and Trif signalling pathways do not really operate independently of each other but are, in fact, interlinked. TNFa production in response to TLR4 signalling, for example, requires both MyD88 and Trif for efficient cytokine production (doi: 10.1016/j.celrep.2022.111225). The differences in LPS chemotypes seen by the authors may simply reflect differences in efficacy at TLR4 rather than anything else especially given the absence of Tram in HEK cells. This would need to be fully explored to make the data presented by the authors compelling.

Response authors: We agree with the reviewer that there is a crosstalk between the MyD88- and TRIF/TRAM-dependent pathways. Furthermore, we concur that variations in LPS chemotypes may reflect differing efficacies. However, we believe that these differences in efficacy do not preclude the possibility that LPS from different origins may trigger distinct events after TLR4 activation. It is plausible that both LPS chemotypes result in an identical receptor conformation, leading to the activation of the same pathway, but with differing maximal effects. Nevertheless, structural variations on LPS may give rise to distinct receptor conformations, potentially producing different outcomes. To confirm and quantify a true ligand bias, we see that several questions must first be addressed. However, we believe this assay offers a valuable opportunity to detect events that may be overlooked by conventional methods. We have clarified this in the revised manuscript by rephrasing sections of the results: *“LPS from different sources triggers distinct inflammatory responses and shows a form of biased signaling¹⁵. To test whether label-free, cell-based assays using optical biosensor technology can reliably study TLR signaling in cells expressing endogenous levels of TLRs and detect potential biased signaling, we used THP-1 cells, which can activate both TLR4 signaling pathways.”*

and in the discussion:

“We demonstrate that different LPS chemotypes induce distinct DMR responses, suggesting the possibility of biased signaling downstream of TLR4⁶⁸.”

10. “We transfected TLR4-HEK293 cells with Venus-tagged MyD88 and assessed cellular localization of MyD88 by immunofluorescence analysis”

Data from these experiments will not reflect what is normally happening to MyD88 in cells because this is now an over expression system (HEKs have endogenous MyD88) and MyD88 will oligomerise and now form complexes in a diverse array of locations within the cell. These experiments would need to be performed in a MyD88 knock out back ground and carefully controlled such that the expression levels of tagged MyD88 were similar to those seen endogenously.

We thank the reviewer for highlighting this point and providing suggestions. We have performed additional experiments with HEK293 MyD88-KO cells and provide this new data in Fig. 7c^{NEW} and Supplementary Fig. 13b-f^{NEW}:

“In the absence of LPS, MyD88 was located in condensed form in the cytoplasm, as previously demonstrated⁴⁹. However, after LPS stimulation, MyD88 redistributed throughout the cell (Supplementary Fig. 13a). To determine if this condensation was due to MyD88 overexpression, we generated HEK293 MyD88-KO cells, which lack functional TLR4 activity (Supplementary Fig. 13b-d). HEK293 MyD88-KO cells transfected with TLR4 and endogenous amount of MyD88 Venus (Supplementary Fig. 13e) showed a similar condensation pattern of MyD88 in the cytoplasm (Fig. 7c). After stimulation with LPS E. coli, MyD88-Venus readily dispersed from the condensed structures and formed puncta within 5 minutes. In contrast, LPS S. minnesota induced a delayed formation of MyD88-Venus puncta (Fig. 7c, white arrows). By 45 minutes, the formation of puncta appeared complete for both LPS chemotypes. MyD88-transfected HEK293 MyD88-KO lacking TLR4 showed MyD88 in condensed form also in the presence of LPS (Supplementary Fig. 13f). The differences in the kinetics of MyD88 assembly induced by LPS E. coli and S. minnesota might correspond to the signal traces observed by the optical biosensor (Fig. 7d).”

Reviewer #2 (Remarks to the Author):

I hereby identify myself as Jon Kagan (Harvard Medical School).

In this manuscript, the authors report the use of an optical biosensor assay to study TLR signaling pathways. A significant dataset is presented, which revealed the utility of this assay to discern cellular responses induced by distinct TLR ligands--in distinct cell types. I commend the authors on their creativity and use of this assay, which may become a standard tool for the field.

Response authors: We are honoured to receive such positive feedback and thank Dr Kagan for his highly encouraging comments.

I have a general question of the molecular underpinnings of what optical biosensor assays are actually reporting. The authors could be asked to provide such data, but I would be fine if they save such analysis for a future study. My only specific query is listed below.

Response authors: We now show that the optical biosensor reports cytoskeleton rearrangement (Fig. 2^{NEW} and Supplemental Fig. 4^{NEW}). For further details, please refer to Reviewer 1, comment 1.

1. Line 135 describes Mal as a Myd88-independent regulator of TLR signaling. This statement runs counter to many studies and should be reconsidered.

Response authors: Thank you for spotting this. We have rephrased the sentence:

“These findings offer evidence to suggest that both agonists trigger differences in the recruitment of adaptor proteins, including MyD88 (Myeloid differentiation primary response 88) and Mal (Myd88 adaptor-like protein)³⁴ as well as signal cascade kinetics.

Reviewer #3 (Remarks to the Author):

Holze et al. NatComm:

The manuscript by Holze et al. shows a nice study on Toll-like receptor signaling using a label-free method with the help of dynamic mass redistribution (DMR). The work shows and illustrates the importance of the dynamic and kinetic examination of ligand-induced signaling in the cell. DMR is an innovative and sensitive method for observing signaling processes in the cell that does not require any changes to the cell or the target protein in order to obtain a readout. This work opens up new possibilities for the characterization of the under-researched group of Toll-like receptors and the possible future development of drugs that target TLRs and is therefore significant for the field. The methodology is sound. Even if the work still needs to be improved in some areas, I would like to recommend it for publication in Nature Communications after addressing these issues (major revision).

Response authors: Thank you very much for the thoughtful comments and valuable suggestions. We are grateful for the interest in our work.

I would like to mention the following points:

- I would like to see a bit more specific background on TLRs in the introduction. How many TLRs are known? Can they be classified? Which ones are covered in this study and why? Is there a link to possible diseases (there is talk of possible future drug development later on)?

Response authors: As suggested, we have added more information on TLRs in the introduction:

“The TLR family consists of 10 members (TLR1-TLR10) in humans and 12 members (TLR1-TLR9, TLR11-TLR13) in mice and is known to detect a variety of pathogen-associated molecular patterns (PAMPs) of invading microorganisms (e.g. lipids, lipoproteins, proteins, and nucleic acids) that enable the host to distinguish between different infections³. In addition to PAMPs, also stress- or injury-released molecules, termed damage- or danger-associated molecular patterns (DAMPs), can be sensed by TLRs. They can be broadly classified into two subfamilies based on their localization: cell surface TLRs (TLR1, 2, 4-6, 10) and intracellular TLRs (TLR3, 7-9, 11-13). Mechanistically, TLRs dimerize as homo- or heterodimers for activation, and trigger intracellular pathways that promote diverse cellular responses, including inflammatory processes⁴. Dysregulated TLR signaling has been implicated in various inflammatory and autoimmune diseases, including sepsis, rheumatoid arthritis, cancer, metabolic disorders, and neurodegenerative diseases. Understanding the roles of TLRs in these conditions is essential

for developing targeted therapies that can modulate immune responses, leading to improved treatment outcomes.”

- I am missing chemical structures for the TLR ligands used in this study. Here I would suggest a figure to be placed in the Supporting Information. The small molecule modulators should definitely be shown. Are there specific structures for the LPS and PamXCSK4 ligands? Where exactly are the structural differences? What’s the chemical difference of LPS smooth (s), rough (r)?

Response authors: We have added the schematic structures of LPS chemotypes (Supplemental Fig. 1^{NEW}) and TLR ligands and inhibitors (Supplemental Fig. 15^{NEW}).

- Page 3, line 73: LPS consists of ... (no comma).

Response authors: DONE.

- I am torn with the specification of the DMR readout in seconds. From my point of view, a presentation and discussion in minutes would have been the better choice, as the reader (at least me) finds it difficult to imagine an exact period of time with, for example, 15,000 seconds. In addition, the authors describe their results in seconds, sometimes in minutes and occasionally in hours. This is not conducive to comparability. This should be handled consistently. I would prefer minutes, even if I am aware that this means that a number of figures also have to be adjusted in the x-axis. It should be done consistently in any case. The authors can think about the changeover to minutes (not a major issue).

Response authors: DONE, we now present and discuss the DMR readouts in minutes.

- I think it's really good that the limitations of the method are discussed at the end (black box readout, need of a ligand toolbox and prefindings from other assays, complementary to conventional assays). In my view, it will take a lot of preliminary work on conventional assays and selective ligands to get to a point where DMR can be used as a label-free method and all the advantages mentioned can be utilized. Perhaps this can be added to the existing section. Perhaps at this point the authors can also better emphasize what contribution the method can make to other assays.

Response authors: We thank the reviewer for the positive comment. As suggested, we have included new statements in the discussion:

“Significant preliminary work with conventional assays and selective ligands is still required before optical biosensors can be fully used as a label-free method, allowing it to realize its advantages.

[...]

Our study highlights the potential of this methodology to discover novel activation mechanisms in real time, within living cells, and without the need for tags. This could provide a new dimension for understanding TLR biology and pharmacology.”

- What techniques have been used to investigate TLP signaling pathways to date?

Response authors: We have added more background information in the introduction:

“These studies typically focus on a limited range of downstream signaling pathways and effector proteins. However, commonly used measurements, such as cytokine release and protein phosphorylation, often provide delayed insights. Their regulation by multiple transcription factors and other proteins complicates real-time evaluation of activation. Interpreting these downstream responses is difficult because they occur long after the initial TLR activation events and are susceptible to signal amplification.”

- Page 34, line 812 (Fig.1): “TAK-242” not “TAK.242”.

Response authors: DONE.

- Page 5: The authors discuss previous observations of heterodimers. What assay was used and what is novel in this experiment?

Response authors: The dimerization of TLRs is a fundamental concept in TLR research, and has been demonstrated through various techniques, including crystal structure analysis, confocal FRET microscopy, single-molecule localization microscopy, and co-immunoprecipitation, using TLR-KO cells and receptor mutants. For more detailed information, we refer to previous publications, such as Ozinsky *et al.* PNAS 2000, doi:10.1073/pnas.250476497; Visintin *et al.* J Immunol 2005, doi: 10.4049/jimmunol.175.10.6465; Kim *et al.* Cell 2007, doi: 10.1016/j.cell.2007.08.002; Jin *et al.* Cell 2007, doi: 10.1016/j.cell.2007.09.008; Park *et al.* Nature 2009, doi: 10.1038/nature07830; Kang *et al.* Immunity 2009, doi: 10.1016/j.immuni.2009.09.018; Krüger *et al.* Sci Signal 2017, doi: 10.1126/scisignal.aan1308.

While the optical biosensor does not detect TLR homodimerization (e.g., TLR4) or TLR2 heterodimerization, we demonstrate in HEK293 TLR-KO cells that TLR2 heterodimers can induce potential ligand-dependent biased signaling. Although the underlying mechanisms remain unclear, it is evident that cellular activation following TLR2 ligation depends on two key factors: (i) the expression of additional TLRs in the cell and (ii) the nature of the TLR2 ligand.

- How has dimerization been demonstrated in the literature? Was evidence of dimerization provided in this study?

Response authors: We kindly refer to our previous answer to this question (see above).

- Figure 1/2 caption: I assume the dashed lines represent the six time points that were used to generate CRCs. If that’s true, then please state this in the respective Figure captions.

Response authors: We have added the information in the respective figure captions.

- Page 5, line 120: “while Pam2CSK4 mainly activated TLR2/6”. It should be mentioned here that Pam2CSK4 also triggers significant activation at TLR2/1. Especially as this is discussed in the next chapter (line 155-157). In addition, the designation in Supp. Fig 2a is wrong or at least very confusing. It says “... Pam2CSK4 (TLR 2/6) and Pam3CSK4 (TLR 2/1)”, although the test was performed exactly the other way around as indicated in the brackets. It should also be pointed out in the main text that both Pam ligands were also “cross-tested” at the other dimer target.

Response authors: Thank you for pointing this out. The designation has been corrected (now Supplementary Fig. 5a). The revised manuscript text reads as follows:

“However, both Pam₂CSK₄ and Pam₃CSK₄ were also able to activate the other dimer, although to a lesser extent (Supplementary Fig. 5a).”

- In lines 147-148 (Page 6) it is stated: „We used the well-characterized epidermal HaCaT cell line that responds to TLR2/1 and TLR2/6 ligands, but shows no functional TLR4 signaling [34].“ But in line 155 “HaCaT cells lack the expression of TLR6[34]”. How does that work?

Response authors: Thank you for spotting this flawed statement. We have rephrased the statement:

“We used the well-characterized epidermal HaCaT cell line that responds to TLR2 ligands, but shows no functional TLR4 signaling³⁸.”

- Page 7, Line 162: “... overlapping signals can be excluded”. That might be true for C29 but the signal CU-CPT22 has a negative response after reaching the baseline (Supp. Fig. 3b). Accordingly, overlapping signals in Fig. 3c cannot be ruled out in my opinion.

Response authors: Thank you very much for pointing out this flawed statement. The passage has been rephrased:

“Overlapping signals can be excluded for C29. However, since CU-CPT22 induced negative signals at later time points, interference with Pam₃CSK₄-induced signals may have occurred, independent of TLR2 inhibition.”

- Page 7, Lines 162-163: „The optical traces triggered by the lipopeptides were independent of TLR4 (Supplementary Fig. 3c).“ I do think that the Pam₂CSK₄ signal has been influenced. The course is similar, but the curve as a whole experiences a clear depression, even into the negative response range. How can this be explained?

Response authors: Thank you for spotting this. We agree with the reviewer that Pam₂CSK₄-induced responses have been influenced in the presence of the TLR4 antagonist TAK-242. We have revised the sentence:

“The optical traces triggered by the lipopeptides were largely independent of TLR4 (Supplementary Fig. 6c). The Pam₂CSK₄-induced signals shifted to more negative values in the presence of TAK-242, without altering the overall course of the signals.”

The reason for this is unclear but may be explained by the possible existence of TLR2/TLR4 heterodimers, which have been described in the literature (Zewinger *et al.* Nat Immunol 2020, doi: 10.1038/s41590-019-0548-1). HaCaT show no functional TLR4 signaling but express TLR4 (Ref 38, Köllisch *et al.* Immunology 2005, doi: 10.1111/j.1365-2567.2005.02122.x), thus TLR2/TLR4 heterodimers may partially contribute Pam₂CSK₄-induced signaling. However, whether and these potential heterodimers induce signaling remains unknown and requires further investigation to precisely determine their role.

- Page 8, line 184-185: Please do not abbreviate “RNA-seq”. At least describe it at first appearance.

Response authors: DONE. We now use “RNA sequencing” throughout the manuscript.

- Common question: How can negative peaks be interpreted in DMR readouts compared to positive signals? Is there a direct connection to the mode of action, for example when agonists and inverse agonists are compared?

Response authors: Positive and negative DMR signatures represent an overall increase or decrease in mass near the biosensor. Negative peaks suggest a reduction in the mass of cellular components near the sensor, indicating that these components are moving away from the surface. This can be linked to processes like cell contraction or other events that decrease the density of cellular material close to the sensor. In contrast, positive signals in DMR readouts indicate an increase in cellular mass, such as proteins or cytoskeletal elements, near the sensor surface, often implying cell activation, reorganization, or other events that bring components closer to the sensor. Importantly, DMR signatures cannot be directly correlated with specific signaling pathways activated or inhibited by agonists or antagonists. Rather, they offer insights into cell type-dependent pharmacology, where responses vary based on the cellular context, serving as valuable tools for exploring signaling networks.

- Page 9, line 225: The small molecule ST2825 is described to be a “... synthetic analogue of (protein) MyD88 ...”. It sounds a bit strange to me how a small molecule structure could be an analogue of a protein. Please explain or be more precise. Otherwise this statement/wording is misleading.

Response authors: We have revised the description of the inhibitor:

“We used the MyD88 inhibitor ST2825, a synthetic compound that mimics the structure of the heptapeptide in the BB-loop of the MyD88-TIR domain, thereby interfering with the homodimerization of MyD88, a crucial step for pathway activation⁴⁸ (Fig. 7a).”

- Page 9, line 228: I think the initial negative peak is also abolished by ST2825, isn’t it?

Response authors: Thank you very much for pointing out this flaw. We have corrected the relevant sentence.

- Figure 6b: Indicate the used concentrations (100 ng/mL) as legend instead of “control”.

Response authors: DONE.

- Figure 6b caption: Use spaces between numbers and units! Check throughout the manuscript.

Response authors: DONE.

- I don't think that "blunted" is the right wording for antagonizing the induced effects. This was used at least two times. Please change!

Response authors: DONE. We now use inhibited in both cases.

- Page 13, line 33: I think "that" needs to be deleted (Grammar).

Response authors: DONE.

- Positive: A number of reasonable control experiments were chosen and executed.

Response authors: We thank the reviewer for the kind words.

- Table 1: nH is not described (most likely the Hill slope).

Response authors: DONE, thank you, corrected.

- Table 1/2: I find the specification of the LogEC₅₀ calculated from ng/mL concentrations confusing, not very meaningful and difficult to compare. Are no molecular weights given/known for the compounds so that the concentration could be given in moles?

Response authors: Declaring a precise molecular weight for LPS is challenging due to the inherent variability in its structure (see Supplement Fig. 1^{NEW}). Typically, the molecular weight of LPS ranges from approximately 10,000 to 20,000 Da. Unfortunately, this variability prevents the exact calculation of log EC₅₀ values in moles for LPS. To maintain consistency in the tables, we have also provided the values for Pam₂CSK₄ and Pam₃CSK₄ in ng/ml concentrations.

We thank all the reviewers for the positive feedback.

REVIEWER COMMENTS

Reviewer #1 (Remarks to the Author):

The authors have performed a number of new experiments to address the issues raised by this referee. A few issues still remain, but these can be resolved by important additions to the text as follows:

1. "To verify that the LPS E. coli signal detected in suspension HEK293 reporter cells, which are stably transfected to express TLR4, is mediated by TLR4, the inhibitor TAK-242 was used (Fig. 2a)." Were cells transfected with just TLR4 or TLR4/MD2? Without MD2 LPS cannot signal.

Response authors: The HEK293 TLR4 reporter cells were stably transfected to express TLR4, MD-2 and CD14. We have rephrased the sentence:

"To verify that the LPS E. coli signal detected in suspension HEK293 TLR4/MD-2/CD14 reporter cells is mediated by TLR4, the inhibitor TAK-242 was used (Fig. 2a)."

2. Were the experiments with THP-1 cells done in the presence of serum? If so the serum will provide soluble CD14. I don't think CD14 is necessarily important for the MS, but the text must be correct so even if the cells don't appear to express CD14, the serum must be taken into account and a comment added accordingly to the text.

Response authors: We thank the reviewer for pointing this out. The experiments with THP-1 monocytes were done under serum-free conditions. We have added a detailed description in the methods section and rephrased a sentence in the results section:

"Given that THP-1 monocytes express little⁴⁴ to no CD14 (Supplementary Fig. 7) and were stimulated under serum-free conditions, our findings are in line with these observations."

3. Fig 7: The m-venus MyD88 aggregates in HEKs are not present in fluorescently tagged MyD88 in macrophages. This suggests the HEK model is a problem. In the opinion of this reviewer data with the m-Venus MyD88 should be removed from the MS. If the authors think it is critical to have these data in the MS then a sentence needs to be added stating that "whilst aggregated tagged MyD88 is what is seen in HEK cells this does not reflect the behaviour of MyD88 under physiological conditions for example in macrophages"

Response authors: We agree with the reviewer that the aggregated tagged MyD88 is an artefact and non-physiological. However, we believe that the data with Venus-tagged MyD88 may at least partially correspond to the signal traces observed by the optical biosensor. Therefore, they are important in the interpretation of the DMR signatures. In response to the reviewer's suggestion, we have rephrased a sentence in the results section:

"To determine whether this condensation, which does not reflect the behaviour of MyD88 under physiological conditions, was due to MyD88 overexpression, we generated HEK293 MyD88-KO cells, which lack functional TLR4 activity (Supplementary Fig. 13b-d)."